



# Seasonal effects in the long-term correction of short-term wind measurements using reanalysis data

Alexander Basse[1,2], Doron Callies[1,2], Anselm Grötzner[3], and Lukas Pauscher[2]

[1]Department of Integrated Energy Systems, University of Kassel, Wilhelmshöher Allee 73, 34121 Kassel, Germany
[2]Fraunhofer Institute for Energy Economics and Energy System Technology (IEE), Königstor 59, 34119 Kassel, Germany
[3]Ramboll Deutschland GmbH, Elisabeth-Consbruch-Straße 3, 34131 Kassel, Germany

**Correspondence:** Alexander Basse (alexander.basse@uni-kassel.de)

**Abstract.**

Measure-Correlate-Predict (MCP) approaches are often used to correct wind measurements to the long-term wind conditions on site. This paper investigates systematic errors in MCP-based long-term corrections which occur if the measurement on site covers only a few months (seasonal biases). In this context, two common linear MCP methods are tested and compared, namely Variance Ratio and Linear Regression with Residuals. Wind measurement data from 18 sites with different terrain complexity in Germany are used (measurement heights between 100 and 140 m). Six different reanalysis data sets serve as the reference (long-term) wind data in the MCP calculations. Besides experimental results, theoretical considerations are presented which provide the mathematical background for understanding the observations. General relationships are derived which trace the seasonal biases to the mechanics of the methods and the properties of the reanalysis data sets. This allows the transfer of the results of this study to different measurement durations, other reference data sets and other regions of the world. In this context, it is shown both theoretically and experimentally that the results do not only depend on the selected reference data set but also significantly change with the choice of the MCP method.

## 1 Introduction

An extensive measurement campaign generally constitutes an essential part of wind resource assessment and, therefore, of a successful wind energy project. In most cases, these measurements provide around one year of wind data at the site of interest (Lackner et al., 2008). Inter-annual variations in wind speed are reported to vary by between 4 % and up to 10 % (e.g., Corotis, 1976; Justus et al., 1979; Klink, 2002), depending on the respective site; hence, the measured wind data usually do not represent the long-term wind conditions. This aspect becomes even more momentous when the energy in the wind is considered which has been reported to vary by 6 % (Pryor et al., 2018) up to 20 or even 30 % (Albrecht and Klesitz, 2006; Pryor et al., 2006; Corotis, 1976) from year to year. To account for this issue, a long-term correction is performed.

For this purpose, reference data are needed which should be available for a long-term period of one to two decades (Carta et al., 2013; Liléo et al., 2013; Lackner et al., 2008) and show a high degree of similarity to the measured wind data (e.g., a high correlation coefficient of measured and reference data).





Over the recent past, reanalysis data gained more and more popularity in the wind industry and are now used extensively in
wind resource assessment (Ramon et al., 2019; Miguel et al., 2019). This type of data is characterized by a combination of both
meteorological observations as well as numerical models which simulate the state of the atmosphere (climate and weather).
Different types of reanalysis data are available, ranging from (often freely available) global data sets (e.g., MERRA-2 by NASA
(NASA, 2019), ERA5 by ECMWF (CDS, 2018)) to mesoscale reanalyses which generally are not free of charge but provide
higher spatial resolution.

A statistical procedure relating the reference data to the measured data is performed to derive a correction function. In this
context Measure-Correlate-Predict (MCP) approaches have evolved to become a standard tool for wind farm developers (Carta
et al., 2013). These methods model a statistical relationship between the time series of the reference and the measurement
data. Afterwards, the relationship is applied to the long-term reference data providing the long-term wind conditions. The
relationship between reference and target data, therefore, is assumed not to be time-dependent, i.e., valid in the correlation
period as well as in the correction period.

Numerous MCP methods are used in modern wind resource assessment applications. They range from simple linear models
(e.g., Rogers et al., 2005a; Weekes and Tomlin, 2014a; Romo Perea et al., 2011; García-Rojo, 2004) to complex machine
learning approaches like neural networks (e.g., Albrecht and Klesitz, 2006; Velázquez et al., 2011; Jie Zhang et al., 2014; Bass
et al., 2000; Bilgili et al., 2007). The investigation and comparison of different MCP approaches has been subject to a large
amount of scientific publications. In Carta et al. (2013) an extensive review is given on existing MCP methods applied in wind
resource assessment and related research fields. It is concluded that, by far, the most commonly used MCP methods in the
wind industry are based on linear approaches. Other studies confirm this observation and underline the benefit of the simplicity
of linear MCP methods for use in wind energy applications (e.g., Weekes et al., 2015; Weekes and Tomlin, 2014c; Sørensen
et al., 2011). In a Round-Robin experiment in Germany in 2018 it was found that 24 of 29 consultants used linear correlation
methods which mostly overperformed more complicated approaches (Basse et al., 2018).

In order to enable a precise determination of the relationship between measurement and reference data, a sufficient amount
of measurement data is necessary, that is, the concurrent period needs to be "long enough". Various studies have been presented
which address the question of how long the time span covered by the measurement should be. In general, it is recommended
to be at least one year (Carta et al., 2013), while the use of complete years is important as an uneven representation of different
months increases the uncertainty (Liléo et al., 2013; Taylor et al., 2004). As a consequence of such studies, an amount of 12
months of measurement is recommended or even a mandatory minimum duration due to technical guidelines and standards
such as FGW e.V. (2020), IEC (2017) or MEASNET (2016).

From an economic perspective, though, there is a strong desire to reduce the duration of the measurement in order to save time
and money (Carta et al., 2013). This is especially true with the increasing popularity of lidar measurements, which have a high
mobility and low installation costs compared to classical measurement masts. Moreover, an estimate of the wind conditions on
site often is of interest for the wind park planner before the measurement campaign is completed. In all such cases, a smaller
amount of wind data needs to be dealt with and a long-term correction is performed based on wind measurement data which
comprise much less than a year.





However, seasonal effects occur when the measurement does not cover all seasons (Rogers et al., 2005a; Weekes and Tomlin, 2014a,b,c; Saarnak et al., 2014) resulting in a dependence of the estimated energy yield on the period in which the measurement is conducted. These can induce systematic deviations and, thus, increase the uncertainty of the resource assessment significantly. Therefore, understanding seasonal patterns in long-term correction and their relation to data sources and the choice of the MCP method is of high interest for the wind industry.

Several studies have investigated the accuracy of a long-term correction (LTC) of short-term wind measurements in dependence of the measurement duration (e.g., Rogers et al., 2005a,b; Weekes and Tomlin, 2014c; Weekes et al., 2015; Taylor et al., 2004; Romo Perea et al., 2011; Miguel et al., 2019). While in some of these, seasonal effects are broadly addressed, to the authors' knowledge there is a lack of scientific publications which give profound explanations for seasonal patterns in biases of the LTC.

This paper investigates seasonal effects and related biases in wind speed (mean and variance) and annual energy yield in the LTC induced by short (three months) measurement periods. Motivated by their relevance for practical use, two linear MCP methods are applied and compared: Linear Regression with Residuals (Weekes and Tomlin, 2014a) and the Variance Ratio method (Rogers et al., 2005a). First, theoretical considerations are developed to assess the impact of varying statistical relationships between the measurement and the reference data in the short-term period when compared to the long-term period. In a second step, wind measurement data from 18 sites in Germany and six different reanalysis data sets are used to assess the significance and magnitude of seasonal effects in the LTC. Interrelations of the seasonal effects with properties of the reference data and the correlation method are analyzed both theoretically and experimentally.

## 2 Measurement and reanalysis data used in this study

An overview of the measurement campaigns is given in Tab. 1. All sites are located in Germany; the complexity of the sites ranges from flat agricultural areas to the hilly low mountain ranges in Central Germany (one of the complex sites is described in Pauscher et al. (2018)). For all sites a time series of an entire year for a height level between 100 and 140 m is available, representing typical hub heights of modern wind turbines. The data were collected by profiling lidar (Light Detection And Ranging, see e.g., Emeis et al. (2007)) of type Leosphere WindCube V1 & V2 (Leleu, 2019), sodar (Sound Detection And Ranging, see e.g., Bradley (2008)) or mast measurements.

Measurement data were collected at a temporal resolution of 10 minutes and then averaged to hourly values (centered at the full hour) to comply with the typical temporal resolution of the reanalysis data (see below). The availability of the measurement data is higher than 80 % at all sites with more than 90 % data availability at 14 sites. All data gaps are smaller than 100 consecutive hours except for a single site (Site 17 in Tab. 1), where approx. 10 days of data are missing in winter (overall data availability for this site: 95 %).



**Table 1.** Details of the measurement sites. The duration of the individual measurements is exactly one year. The measurements were carried out between May 2013 and April 2019.

| Site No. | orography and surface cover | measurement height [m] | measurement device |
|---|---|---|---|
| 1 | hilly, forested | 140 | Lidar (WindCube V2) |
| 2 | slightly hilly, forested | 140 | Lidar (WindCube V2) |
| 3 | mainly flat, forested | 140 | Lidar (WindCube V2) |
| 4 | hilly, sparsely forested | 140 | Lidar (WindCube V1) |
| 5 | slightly hilly, barely forested | 140 | Lidar (WindCube V1) |
| 6 | slightly hilly, forested | 140 | Lidar (WindCube V2) |
| 7 | hilly, forested | 140 | Lidar (WindCube V1) |
| 8 | slightly hilly, no trees | 140 | Lidar (WindCube V1) |
| 9 | slightly hilly, sparsely forested | 140 | Lidar (WindCube V1) |
| 10 | mainly flat, buildings nearby | 135 | Lidar (WindCube V2) |
| 11 | mainly flat, small town nearby | 140 | Lidar (WindCube V2) |
| 12 | hilly, forested | 135 | Mast |
| 13 | slightly hilly, forested | 140 | Mast |
| 14 | rather flat, forested | 130 | Mast |
| 15 | flat, close to a city | 110 | Mast |
| 16 | flat, agricultural area | 100 | Mast |
| 17 | rather flat, forest nearby | 140 | Sodar |
| 18 | slightly hilly, forested | 140 | Sodar |

The following six different reanalysis data sets serve as reference data in the MCP calculations:

1. **MERRA-2** (GMAO, 2015). The Modern-Era Retrospective Analysis for Research and Applications Version 2 (MERRA-2) is based on global numerical weather analyzes of the U.S. National Aeronautics and Space Agency (NASA). The data are available as one-hour time series since 1980 for a height of 50 m and a spatial resolution of 0.5° x 0.66°. The time stamps refer to average hourly values centered at 00:30 h, 01:30 h etc. In order to obtain comparability with the other reanalysis data sets and consistency in temporal terms, these were interpolated to values centered at the full hour.

2. **ERA5** (CDS, 2018). The data set is calculated at the European Centre for Medium-Range Weather Forecasts (ECMWF) and provided by the Copernicus Climate Change Service. The ERA5 data represent the follow-up data set to the ERA-Interim reanalyses of the ECMWF. The spatial resolution of the ERA5 data is approx. 31 km ($\approx 0.28°$). Long-term series of this data set are available for 100 m above ground in an hourly resolution. In contrast to the MERRA-2 data, these data are instantaneous values instead of averaged wind speeds (centered at the full hour).

3. **EMD-ConWx** (EMD, 2020a). This data set is created using the WRF model (Weather Research & Forecasting Model, see WRF (2020)) and is provided by EMD International A/S from Denmark. It is based on the ERA-Interim reanal-



ysis data of the ECMWF, refined to a resolution of 3 km. The temporal resolution of the long-term time series is 1 h (instantaneous values centered at the full hour). Wind data are provided at heights of 10, 25, 50, 75, 100, 150, and 200 m.

4. **EMD-WRF Europe+** (EMD, 2020b). This dataset is a further development of the EMD-ConWx data. The ERA5 re-analysis data have replaced the ERA-Interim data, while spatial resolution and temporal properties did not change. Wind data are provided at the same heights as in EMD-ConWx and six additional heights up to 4000 m.

5. **anemosM2:** anemos Windatlas based on MERRA-2 (anemos, 2020a,c). Similar to the EMD data sets, these data are created based on a downscaling of global reanalysis data (here: MERRA-2) using the WRF model (version 3.7.1) to a resolution of 3 km. In contrast to the other models, anemos uses statistical post-processing based on measurement data, known as remodeling, to improve the simulation results. Furthermore, additional downscaling of the data from the 3 km grid to the specific site is applied. The heights of the wind data generally are freely selectable between 40 and 200 m; for the analysis in this study, wind data at 100 and 140 m were provided.

6. **anemosE5:** anemos Windatlas based on ERA5 (anemos, 2020b,c). This data set is similar to the anemosM2 but uses ERA5 data. Furthermore, in the course of the remodeling, a seasonal correction is performed, i.e., biases in the seasonal course of the ERA5 data are corrected before the statistical downscaling is implemented. The goal is to better capture the seasonal behaviour of the wind conditions. Additionally, a more precise consideration of the roughness at the respective site represents a further difference to the anemosM2 data. Both the magnitude of the seasonal corrections as well as the modifications on roughness constitute a trade secret of anemos (anemos, 2021).

It should be noted that both the anemosM2 and anemosE5 models generally provide a temporal resolution of 10 minutes. In order to guarantee comparability of the results, these were averaged to 1 h ensuring the same temporal resolution for all reanalysis data sets.

In general, reanalysis data are modeled for different locations on a geographical grid. In this study, data were selected from the grid point closest to the respective site. For data sets 3. - 6. data at more than one height level were provided. In these cases, the data at the height closest to the measurement were used (i.e., 100 and 150 m for EMD-ConWx and Emd-Wrf Europe+, 100 and 140 m for the two anemos data sets). For the MERRA-2 and ERA5 data sets the data at the given height (i.e., 50 and 100 m, respectively) were used, i.e., no vertical extrapolation (or interpolation) was performed in this study.

## 3 Methodology

This study compares statistics as observed over different periods in the investigated data - namely short-term data and long-term data. For this purpose, the convention is applied that capital letters are used for long-term variables (e.g., the long-term corrected wind speed) while parameters in lower case letters represent data from the short-term period. The indices $meas$, $ref$, and $corr$ refer to measurement, reference (i.e., reanalysis) and corrected data, respectively.



### 3.1 Selection of short-term periods and procedure of long-term correction

Short-term periods with a duration of 90 days are selected starting at the first day of year and running through the data with an increment of three days ("sliding window"). When the end of the data is reached, the data from the beginning of the year is appended ensuring that 122 90-days measurement periods can be investigated for all reanalysis data sets and all sites.

In a first step, the data in these three-month data portions are analyzed with respect to, e.g., mean and variance of wind speed (Sect. 5.1, 5.2 and 5.3). Secondly, MCP predictions are performed. Regression parameters are derived using the short-term data and, afterwards, correction is performed in the entire one-year period in which measurement data are available. Finally, the corrected data are compared to the measured one-year data (benchmark) and error scores are derived (see Sect. 3.3).

The results, therefore, do not represent the overall errors (or uncertainty) of an LTC in general, which usually is performed over a period of ten years or more (Liléo et al., 2013; Lackner et al., 2008; Carta et al., 2013). Instead, the analysis provides findings on systematic errors (seasonal biases) which emerge due to the reduction of the measurement duration from one year to three months.

It should be noted that in practical applications often a sector-wise regression is performed for an LTC of measurement data comprising a whole year. This means, that the regression parameters are calculated separately for different wind direction bins which allows to take the effects of terrain on wind flow into account. This can be important especially in a complex environment (López et al., 2008). For the shorter three-month periods sectorwise binning, however, generally yielded slightly worse results in this study (presumably due to low data coverage in the different direction sectors). This procedure is, therefore, not applied here. It is acknowledged, though, that in some specific cases a sectorwise approach can be a reasonable choice for an LTC of short-term measurements nevertheless.

When correction is performed, few negative wind speed values can occur. In this study, these values were set to zero.

### 3.2 Long-term correction: Measure-Correlate-Predict (MCP) approaches

In this section, a short overview of the two MCP methods is given which are used in this study. Both implement a linear model to derive a relation between measurement and reference wind speed ($u_{meas}$ and $u_{ref}$). This linear relationship generally is expressed in the form

$$u_{meas} = \beta_0 + \beta_1 \cdot u_{ref} + \varepsilon, \tag{1}$$

where $\beta_0$ and $\beta_1$ represent the main regression parameters. $\varepsilon$ indicates the residuals (deviations from data points to fitting line, see e.g., Ellison et al. (2009)).



### 3.2.1 Linear regression with residuals

160 The probably most widely used linear model is simple linear regression. In this approach the respective regression parameters $\beta_{0,LR}$ and $\beta_{1,LR}$ are calculated via the linear least squares method which minimizes the average squared deviation of the data points from the fitting line (see e.g., Draper and Smith, 1998). This results in

$$\beta_{1,LR} = r_{ref,meas} \cdot \frac{\sigma_{meas}}{\sigma_{ref}} \tag{2}$$

and

165
$$\beta_{0,LR} = \bar{u}_{meas} - \beta_{1,LR} \cdot \bar{u}_{ref}, \tag{3}$$

where $\sigma_{meas}$ and $\sigma_{ref}$ represent the standard deviation of reference and measurement data in the measurement period, and $r_{ref,meas}$ the Pearson correlation coefficient of the respective data. The bar denotes the mean. In the correction period, the relationship is applied to each of the time-series values of the reference data $U_{ref}$ yielding the corrected wind speed values $U_{corr}$:

170
$$U_{corr} = \beta_{0,LR} + \beta_{1,LR} \cdot U_{ref}. \tag{4}$$

A disadvantage of this model is that the variance of the corrected data $u_{corr}$ is reduced in comparison to the measured data $u_{meas}$:

$$
\begin{aligned}
Var(u_{corr}) &= \beta_{1,LR}^2 \cdot Var(u_{ref}) \\
&= r_{ref,meas}^2 \cdot \frac{\sigma_{meas}^2}{\sigma_{ref}^2} \cdot Var(u_{ref}) \\
&= r_{ref,meas}^2 \cdot Var(u_{meas})
\end{aligned}
\tag{5}
$$

This yields $Var(u_{corr}) < Var(u_{meas})$ as, in practical applications, the correlation coefficient $r_{ref,meas} < 1$. Therefore,
175 simple linear regression can be considered a method which generally yields accurate mean wind speeds (Romo Perea et al., 2011; Jie Zhang et al., 2014; Weekes and Tomlin, 2014a; Rogers et al., 2005a; Bass et al., 2000) but not accurate variances; hence, biased estimates of wind speed distribution and energy production can be expected.

A model which addresses this shortcoming and further develops the simple linear regression approach is the *Linear Regression With Residuals* (LR) method discussed in Weekes and Tomlin (2014a). In contrast to simple linear regression, the residuals
180 $\varepsilon$ are explicitly considered giving the missing variance to the corrected data:

$$U_{corr} = \beta_{0,LR} + \beta_{1,LR} \cdot U_{ref} + \varepsilon. \tag{6}$$





$\varepsilon$ is randomly drawn from a normal distribution $\varepsilon \sim \mathcal{N}(\mu = 0, \sigma_\varepsilon)$ with mean $\mu$ and standard deviation $\sigma_\varepsilon$. $\sigma_\varepsilon$ defines the amount of additional scatter and, therefore, inserts small deviations from the fitting line. It is estimated from the data in the measurement period. Weekes and Tomlin (2014a) show that the LR method yields precise mean wind speeds as well as accurate mean wind power densities.

### 3.2.2 Variance Ratio

In Rogers et al. (2005a), the *Variance Ratio* (VR) method as an alternative to the classical linear regression methods is proposed. This approach is closely related to (simple) linear regression; in contrast, however, the regression parameters $\beta_{0,VR}$ and $\beta_{1,VR}$ are not calculated using the linear least square method. Instead, $\beta_{1,VR}$ is defined as

$$\beta_{1,VR} = \frac{\sigma_{meas}}{\sigma_{ref}}. \tag{7}$$

which resembles the particular case of a simple linear regression with correlation coefficient $r_{ref,meas} = 1$ (compare Eq. (2)). This choice of $\beta_{1,VR}$ ensures that the variance is maintained, in terms of equal variances of measured data $u_{meas}$ and corrected data $u_{corr}$ in the measurement period.

$\beta_{0,VR}$ is then computed using Eq. (3) accordingly. This, in turn, ensures that the mean values of measured and corrected data (in the measurement period) are equal. The VR approach therefore maintains both the first and the second order statistical moment of the measured time series in the LTC. Correction is performed via Eq. (4) using the respective regression parameters $\beta_{0,VR}$ and $\beta_{1,VR}$.

In Rogers et al. (2005a) the authors found that the VR method yielded accurate predictions of all investigated metrics including mean wind speed and wind speed distribution. Other studies confirm the suitability of the VR method in the context of long-term correction of wind measurements (see e.g., Weekes and Tomlin, 2014a; Weekes et al., 2015).

### 3.3 Statistical analysis and definition of error scores

For each MCP calculation according to Sect. 3.1, a one-year time series is generated. Based on comparison with the measured one-year data, the following error scores are derived to evaluate the accuracy of these time series:

1. Bias in (annual) mean wind speed, $Err_{mean}$

2. Bias in variance of (one-year) time series, $Err_{var}$

3. Bias in energy density, $Err_{ED}$

   As relative values are addressed only, the bias in energy density is simply based on the bias in cubed wind speed $u^3$ here. The exact procedure of calculation is given in the text of the respective section (Sect. 5.4.3).

4. Bias in theoretical annual energy production of a wind turbine, $Err_{turbine}$





To derive this error score, the theoretical one-year energy production of a wind turbine is calculated using the power curve of a 3.2 MW wind turbine (see Enercon, 2019). This power curve has a cut-in wind speed at 2 m/s and the nominal power is reached at wind speeds of 14 m/s. When the winds are stronger than 25 m/s, no energy is converted (cut-out wind speed). $Err_{turbine}$ is given by the relative deviation of the energy values calculated from the corrected and the measured one-year time series.

## 4   Theoretical considerations


Before experimental analysis is presented, theoretical aspects are discussed. It should be noted that the findings here are, to some extend, also valid for a long-term assessment which is based on an entire year of measurement data. In this case, the inter-annual variations of the wind conditions represent the key factor. However, these are usually smaller than the seasonal variations during the year.

### 4.1   Influence of mean and variance on the estimate of energy


Both mean and variance of the predicted wind speed distribution have an impact on the estimate of the wind power which is, eventually, the target value of a wind resource assessment when planning a wind park. In this section, the importance of an error in each of the two statistical metrics is analyzed.

    It is known that the power in wind $P$ scales with the wind speed in third power ($u^3$). Hence, the expected value $E[P]$ of a
wind speed distribution is mainly characterized by (proportional to) $E[u^3]$. Romo Perea et al. (2011) give an approximation for $E[u^3]$ based on the first three statistical moments of the wind speed distribution,

$$E[u^3] = \bar{u}^3 + 3 \cdot \bar{u} \cdot \sigma_u^2 + \gamma \cdot \sigma_u^3, \tag{8}$$

with $\sigma_u$ representing the sample standard deviation of wind speeds $u$ and $\gamma$ the skewness coefficient. The bar denotes the mean. Generally, $\gamma$ is rather small (Romo Perea et al., 2011) and the term $\gamma \cdot \sigma_u^3$ therefore will be neglected in the following.
Applying the (simplified) formula of the Taylor series method for propagation of error (see e.g., Coleman, 2009),

$$\Delta E[u^3] = \frac{\partial E[u^3]}{\partial \bar{u}} \cdot \Delta \bar{u} + \frac{\partial E[u^3]}{\partial \sigma_u^2} \cdot \Delta \sigma_u^2, \tag{9}$$

with $\Delta$ symbolizing the error of the respective parameter, yields

$$\frac{\Delta E[u^3]}{E[u^3]} = \left(1 + \frac{2}{1 + \frac{3}{A}}\right) \cdot \frac{\Delta \bar{u}}{\bar{u}} + \frac{1}{1 + \frac{A}{3}} \cdot \frac{\Delta \sigma_u^2}{\sigma_u^2} \tag{10}$$

as a formula for the overall relative error of $E[u^3]$. The substitution $A = \bar{u}^2 / \sigma_u^2$ was introduced for means of readability.





The available one-year measurement data (see Sect. 2) were used to derive values for $A$ which are typically present at the
investigated sites. It was found that $A = 5.0 \pm 0.8$ (mean $\pm$ 1 standard deviation). Inserting in Eq. (10) shows that the effect of
a relative error in mean wind speed is weighted six times as strong as the relative error in variance $\sigma_u^2$.

Note that simplifications were applied (e.g., neglection of the skewness of the distribution) and that the output of Eq. (10)
varies from site to site (due to a site-dependence of the parameter $A$). However, a clear impression of a much larger importance
of a high accuracy in mean than in the variance of the wind speed distribution is obtained. As will be shown in the experimental
section (Sect. 5), the errors in variance can be quite large when a long-term correction of short-term wind measurements is
performed and, hence, should not be neglected nevertheless.

Following these considerations, the sections below address the question which factors influence the accuracy of the estima-
tion of the mean and the variance when a long-term correction is performed based on one of the two linear MCP approaches.

## 4.2   Considerations on seasonal bias in mean wind speed

In both cases of the VR and the LR method, the mean value of the corrected wind speed data is given by

$$\bar{U}_{corr} = \beta_0 + \beta_1 \cdot \bar{U}_{ref}, \tag{11}$$

with the respective values of $\beta_0$ and $\beta_1$.

Using the definition of $\beta_0$ (see Eq. (3)) leads to

$$\bar{U}_{corr} = \bar{u}_{meas} - \beta_1 \cdot (\bar{u}_{ref} - \bar{U}_{ref}). \tag{12}$$

The error in mean wind speed usually is defined as the deviation of the calculated mean wind speed from the "true" value.
Hence, the difference $\bar{U}_{corr} - \bar{U}_{meas}$ provides a convenient formula for the theoretical bias in mean wind speed

$$\begin{aligned} Err_{mean,theo} &= \bar{U}_{corr} - \bar{U}_{meas} \\ &= (\bar{u}_{meas} - \bar{U}_{meas}) - \beta_1 \cdot (\bar{u}_{ref} - \bar{U}_{ref}). \end{aligned} \tag{13}$$

This formula is valid for both the LR and VR method (with respective regression parameter $\beta_{1,LR}$ or $\beta_{1,VR}$).


Therefore, three factors have a direct impact on the accuracy in mean wind speed when applying either the VR or LR method:

**(I)  $\bar{u}_{meas} - \bar{U}_{meas}$: Deviation of "true" mean wind conditions (measured data) in measurement and long-term period**

This part of Eq. (13) denotes the difference of mean wind speeds in measurement and long-term period. It, hence, is
a measure for the representativity of the period in which the measurement is carried out. In case of periods of lower





wind speeds, this quantity is negative ($\bar{u}_{meas} < \bar{U}_{meas}$) while positive values occur in case of periods with strong winds ($\bar{u}_{meas} > \bar{U}_{meas}$).

**(II)** $\bar{u}_{ref} - \bar{U}_{ref}$**: Deviation of the mean wind speeds of the reanalysis data in measurement and long-term period**

Similarly to term **(I)** but related to the reanalysis data, this term reflects the differences of wind conditions in measurement and long-term period given by the reanalysis data.

**(III)** **Regression parameter** $\beta_1$

The regression parameter $\beta_1$ weights term **(II)** and, therefore, determines whether the first or the second part of Eq. (13) dominates. As $\beta_1$ is different for the LR and the VR method, the respective results of an LTC will inevitably show differences, accordingly.

Obviously, the value of $Err_{mean,theo}$ is zero when the terms $\bar{u}_{meas} - \bar{U}_{meas}$ and $\beta_1 \cdot (\bar{u}_{ref} - \bar{U}_{ref})$ cancel out. While $\bar{u}_{meas} - \bar{U}_{meas}$ solely depends on the selected measurement period and the specific site, $\bar{u}_{ref} - \bar{U}_{ref}$ is, additionally, highly sensitive to the selected reference data set (reanalysis data in this study) and its capability to reflect the measured seasonal course on site. $\beta_1$, in turn, is dependent on the selected MCP method and can vary in time. In case of representative wind conditions (i.e., small values of terms **(I)** and **(II)**), the exact value of $\beta_1$ is of minor importance.

## 4.3 Considerations on seasonal bias in variance

Similarly to the considerations on mean wind speed above, in this section a theoretical perspective on the accuracy in variance is given. For the variance of the corrected data $Var(U_{corr})$ the following relationship is valid for both VR and LR:

$$Var(U_{corr}) = \beta_1^2 \cdot Var(U_{ref}) \qquad (14)$$

(with the respective values $\beta_{1,VR}$ and $\beta_{1,LR}$ for the VR and LR approach, respectively; cmp. Eq. (5)). As stated above, $\beta_1$ differs for the two MCP methods by the correlation coefficient. For the VR method one obtains

$$Var(U_{corr}) = \frac{Var(u_{meas})}{Var(u_{ref})} \cdot Var(U_{ref}). \qquad (15)$$

The accuracy of the LTC in variance, therefore, directly depends on how the reanalysis data reproduce the "true" variance both in the correlation (measurement) period as well as the long-term correction period. In other words, the ratio of the variances needs to be similar in the correlation and the correction period to yield accurate results.

When the LR method is applied, the respective formula reads:

$$Var(U_{corr}) = r_{ref,meas}^2 \cdot \frac{Var(u_{meas})}{Var(u_{ref})} \cdot Var(U_{ref}) + Var(\varepsilon). \qquad (16)$$





Hence, the variance of the output data is mainly influenced by three factors here:

1. the accuracy of the reanalysis data in reproducing the variance (similarly as discussed for the VR method)

2. the correlation coefficient (in the context of $\beta_{1,LR}$, cmp. Eq. (2))

3. the residuals determined in the measurement period (s. Sect. 3.2.1) and their representativity for the entire correction period

## 5   Experimental Results

In the following sections, the theoretically derived aspects are further explored and tested experimentally. Afterwards, MCP
calculations are presented. Systematic biases are described and discussed. In a last section, the variation of the results between the different sites is explicitly considered.

### 5.1   Annual course of mean wind speed in measurement and reanalysis data

Equation (13) in Sect. 4.2 constitutes the essential basis for the understanding of seasonal biases in mean wind speed in the context of long-term correction of wind measurements. According to that formula, both the annual course of measured wind
speed as well as the capability of the reanalysis data to reproduce this course are decisive.

In Central Europe –the region under investigation in this paper– the wind conditions usually show lower mean wind speeds in summer and stronger winds in winter periods (Pryor et al., 2006). The exact seasonal pattern will be different from site to site, depending on site-related properties (e.g., proximity to sea or topographical conditions). In Fig. 1 the average annual course at the 18 sites as given by the different reanalysis data sets is presented. Additionally, the measured annual course is
shown (black dashed line). In all cases, relative values were used, i.e., the mean wind speeds in the different 90-days periods (see Sect. 3.1) were divided by the annual means of the respective data sets.

All data confirm the typical seasonal pattern described above. Hence, both terms **(I)** and **(II)** in Eq. (13) (i.e., the deviations of the mean wind speeds in short-term and long-term period in measurement or reanalysis data, respectively) will be negative in summer and positive in winter.

For all reanalysis data sets, however, the seasonal course is over-pronounced in comparison to the measured one. In the transitional seasons (spring, fall), the deviations of (relative) reanalysis and measured wind speeds are smallest on average. Differences occur in the amplitudes.

In order to further analyze this aspect, a parameter $d_{mean}$ was calculated aiming to display the deviations from reanalysis to measured data in the seasonal course. $d_{mean}$ is derived based on mean values of reanalysis ($\bar{u}_{ref}$) and measurement data
($\bar{u}_{meas}$) during the 90-days periods in relation to their overall annual mean values ($\bar{U}_{ref}$ and $\bar{U}_{meas}$, respectively):

$$d_{mean} = \frac{\bar{u}_{ref}}{\bar{U}_{ref}} - \frac{\bar{u}_{meas}}{\bar{U}_{meas}}. \tag{17}$$





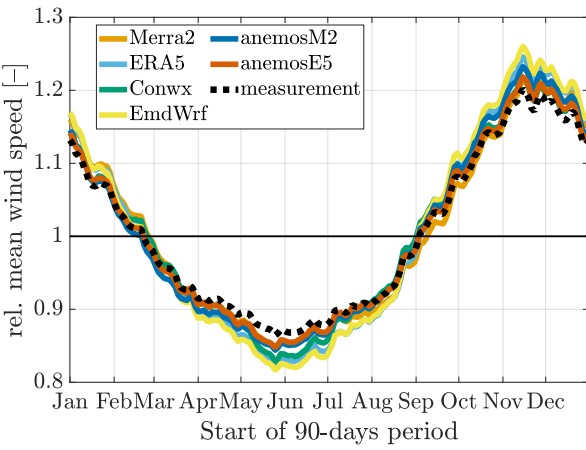

**Figure 1.** Annual course of (normalized) wind speed in reanalysis and measurement data (relative mean values in 90-days periods, arithmetically averaged over all sites).

This quantity, therefore, represents the difference between the colored lines and the measured seasonal course (black line) in Fig. 1. For each short-term period, one value of $d_{mean}$ per site and reanalysis data set is derived. Afterwards, values averaged over all sites are calculated resulting in one set of $d_{mean}$ values for each reanalysis data set.

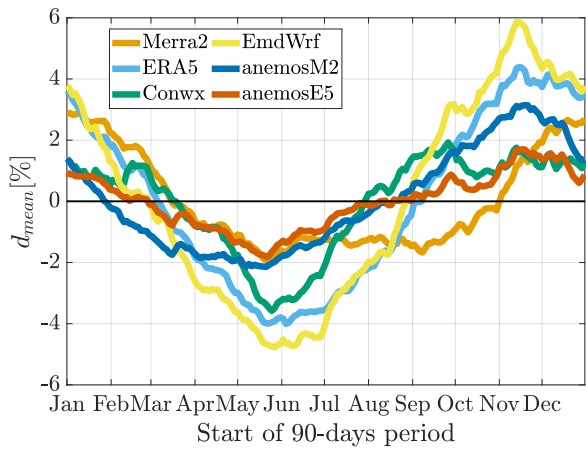

**Figure 2.** Deviation between reanalysis and measurement data in (normalized) mean wind speed (period of 90 days, arithmetically averaged over all sites).

Figure 2 shows the seasonal course of $d_{mean}$. Relatively large differences among the different reanalysis data sets can be observed. The aforementioned over-pronounced seasonal course leads to negative deviations in summer and positive values in winter periods for all reanalysis data sets. Comparing the global reanalysis data sets MERRA-2 and ERA5 with respect





to the accuracy in seasonal course shows advantages for the "older" MERRA-2 data set, as a lower amplitude in Fig. 2 is present. This holds true despite or *because of* the fact that the MERRA-2 data are provided at lower heights (50 m, s. Sect. 2).

This could generally be expected to yield in a lower representativity regarding the seasonal course at the measurement height. However, the ERA5-based anemosE5 data give better results than the MERRA-2 based anemosM2 data. This might be caused by the further developments by anemos when generating the anemosE5 model (e.g., the additional seasonal correction or the remodeling, see Sect. 2).

### 5.2 Seasonal course of regression parameter $\beta_1$

In addition to the aspects regarding the seasonal course of the wind, Eq. (13) underlines that the magnitude of the regression parameter $\beta_1$ plays a significant role. Comparing the respective definitions of $\beta_1$ (Eq. (2) and Eq. (7)) reveals that the VR method always produces larger slopes than the LR method. Fig. 3 (a) and (b) show average regression parameters $\beta_{1,VR}$ and $\beta_{1,LR}$ and their temporal variation during the year. The respective values were calculated during 90-days periods and arithmetically averaged over all sites.

In contrast to $\beta_{1,VR}$, $\beta_{1,LR}$ is subject to clear temporal variations showing lower values in summer and higher values in winter. This, again, reflects the influence of the correlation coefficient which is only considered explicitly in the LR method and which exhibits a seasonal pattern itself (this will be shown in a later section).

According to Eq. (13), the respective $\beta_1$ value weights the seasonal course of the reanalysis data in the determination of the bias in mean wind speed. As a consequence of the findings here, the over-pronounced seasonal cycle of the reanalysis data

as depicted above is weighted stronger in winter than in summer periods when the LR approach is applied. Moreover, lower weighting (in comparison to the VR method) occurs throughout - i.e., $\beta_{1,VR} > \beta_{1,LR}$.

### 5.3 Reproduction of the temporal variation of variance in the reanalysis data

As was shown above, the capability of the reanalysis data in reproducing the variance correctly is decisive for an accurate variance of the generated time series. According to the considerations in Sect. 4.3, this is important in case of both MCP

methods. Therefore, this aspect is briefly addressed here and a measure $d_{var}$ is calculated to investigate this aspect.

Similarly to $d_{mean}$ in Sect. 5.1, $d_{var}$ is defined via the difference of relative values in the 90-days periods,

$$d_{var} = \frac{Var(u_{ref})}{Var(U_{ref})} - \frac{Var(u_{tar})}{Var(U_{tar})}. \tag{18}$$

Figure 4 shows how the temporal variation of the measured variance throughout the year is reproduced by the different reanalysis data sets.

The deviations in variance reach values of up to $\pm$ 10 % and are, therefore, generally higher than the deviations in mean wind speed (see Fig. 2). No universal seasonal dependence can be determined as it was observed for the mean wind speed. Some curves in Fig. 4 show minima in summer and high values in winter or spring while others show contrary characteristics.

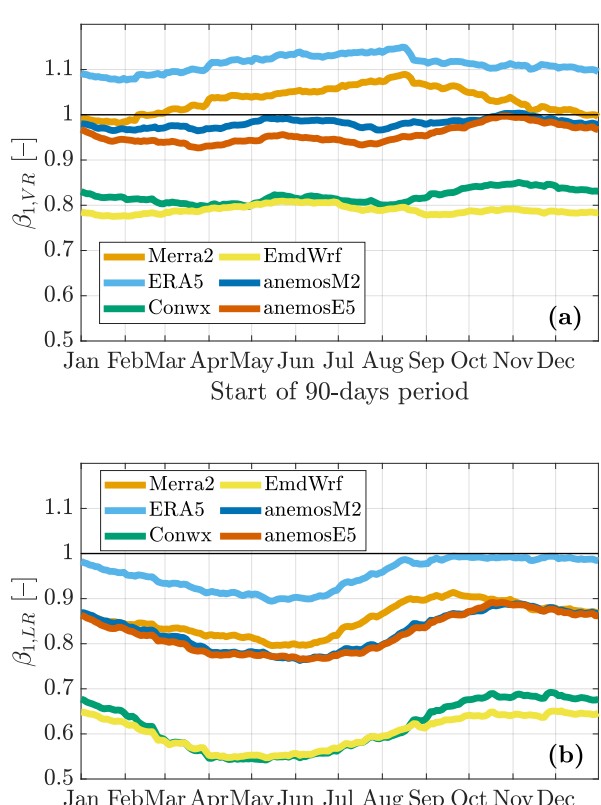

**Figure 3.** Temporal variation of the regression parameter (a) $\beta_{1,VR}$ for the Variance Ratio and (b) $\beta_{1,LR}$ for the Linear Regression with Residuals method. The respective values were determined using a 90-days sliding window and arithmetically averaged over all sites.

### 5.4 MCP calculations: Seasonal bias in mean, variance, and energy

MCP calculations based on 90 days of measurement are now presented. For each reanalysis data set, an average value of
the individual error scores related to one measurement period is calculated by arithmetically averaging over all sites. Due to their importance in the theoretical considerations the focus of the analysis is put on mean and variance of wind speed first. Afterwards, seasonal biases in energy density as well as the (theoretical) energy production of a wind turbine are analyzed. In this context, the influence of the systematic biases in both mean and variance on the accuracy in energy is investigated on an experimental level.

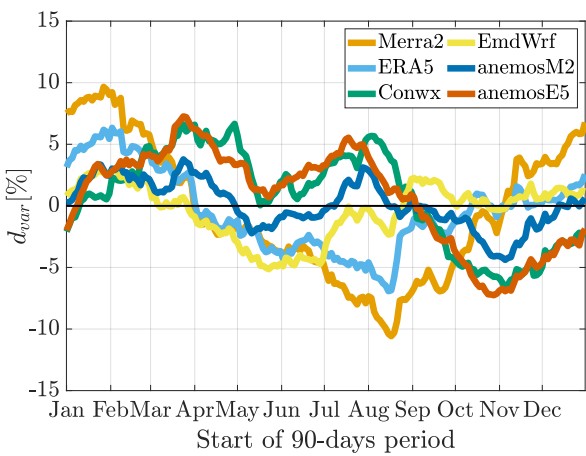

**Figure 4.** Deviation from reanalysis to measurement data in (normalized) variance (period of 90 days, arithmetically averaged over all sites).

### 5.4.1 Seasonal bias in mean wind speed

Figure 5 (a) shows the experimentally obtained error score $Err_{mean}$ using the VR method. An inverse shape to the curves of $d_{mean}$ (i.e., the "error" of the reanalysis data in the seasonal course, see Fig. 2) can be observed: A measurement in summer months results in a positive bias in the corrected wind-speed time series while a negative bias is produced when the measurement is conducted in winter. Thus, a positive bias is produced when the reanalysis data underestimate the (relative) mean wind conditions which prevail in the measurement period and vice versa. These findings are valid for all reanalysis data sets although it should be noted that the shapes of the related curves in $d_{mean}$ are not transformed in the (inverse) course of $Err_{mean}$ in exactly the same way.

Strong differences to these observations and even contrary behaviour can be found when the LR method is used (Fig. 5 (b)). For all reanalysis data sets except ERA5, the mean of the corrected wind speed time series is underestimated in case of measurements in summer, while overestimations prevail for winter measurements. The patterns seem not to be directly related to how the reanalysis data reproduce the measured seasonal course of the mean wind speed. Above that, the ERA5 data gives an inverse curve to all the other reanalysis data sets despite of a high similarity in $d_{mean}$ (Fig. 2). The amplitude of the respective curve is very small indicating a small dependence of the result on the measurement period and, hence, only small seasonal biases.

For most other data sets, the amplitudes of the curves in Fig. 5 (a) and (b) are of comparable magnitude with a slight advantage for the LR method in predicting the mean of the corrected wind-speed time series.

Despite a high similarity in mathematical prospect, the two linear MCP methods yield significantly different results in $Err_{mean}$. The theoretical analysis of the bias in mean wind speed (Sect. 4.3) yielded a theoretical dependence of $Err_{mean}$ on 1.) the representativity of the measurement period for the long-term wind conditions, 2.) connected to that, the similarity of the



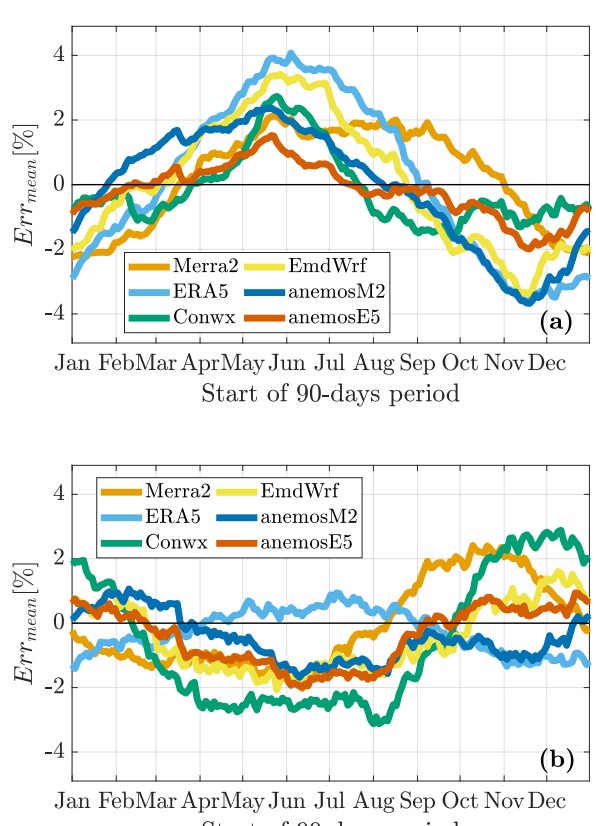

**Figure 5.** Seasonal bias in mean wind speed using the (a) Variance Ratio, (b) Linear Regression with Residuals method.

seasonal course in reanalysis data to the measured one, and 3.) the regression parameter $\beta_1$. As 1.) and 2.) are similar for each reanalysis data set, the differences of the results in Fig. 5 (a) and (b), therefore, must have their reasons in differences in $\beta_1$.

As stated above, the VR method provides larger values here than the LR approach (see Sect. 5.2). This leads to the fact that, generally, the seasonal course of the reanalysis data (term $\bar{u}_{ref} - \bar{U}_{ref}$ in Eq. (13)) is weighted stronger when the VR method is used. As a consequence, the effect of the over-pronounced seasonal course of the reanalysis data (s. Fig. 1 and 2) dominates

here. This is underlined by the fact that $Err_{mean}$ and $d_{mean}$ roughly show inverse shapes. For the LR approach, in contrast, the seasonal course of the reanalysis data is weighted less due to smaller $\beta_{1,LR}$ values. Therefore, in most instances the seasonal pattern measured on site (term $\bar{u}_{meas} - \bar{U}_{meas}$ in Eq. (13)) dominates the overall result. Consequently, most curves of $Err_{mean}$ show a high degree of similarity to the patterns observed in Fig. 1.

As was shown in Fig. 3, in case of the ERA5 data relatively high $\beta_{1,LR}$ values were obtained. For the LR method this causes

a balancing effect (even slightly "overbalanced"). Thus, a relatively small amplitude of $Err_{mean}$ can be observed in Fig. 5 (b)



despite, or rather *because of* the erroneous seasonal course of the ERA5 data. With regard to the VR method, again, highest slopes ($\beta_{1,VR}$ values) were observed for ERA5 compared to the other reanalysis data sets. As a direct consequence, the product of regression parameter $\beta_1$ and (over-pronounced) seasonal course in the reanalysis data clearly dominates the result of Eq. (13) and the highest amplitude can be observed in Fig. 5 (a).

One further example is analyzed briefly here. The largest deviation in the seasonal course $d_{mean}$ was found for the EMD-Wrf Europe+ data set (see Fig. 2). In contrast to the ERA5 data, though, remarkably lower $\beta_{1,LR}$ values are present for this reanalysis data set (see Fig. 3 (b)). Eventually, the product of (small) regression parameter and (large) deviation of the reanalysis data in the seasonal course in Eq. (13) results in a relatively small amplitude of $Err_{mean}$.

     In summary, it can be stated that the capability of the reanalysis data in reproducing the seasonal course of the "true" wind
conditions on site is an important aspect when considering the bias in mean wind speed. However, positive (or negative) deviations in seasonal course do not transform to negative (or positive) biases directly. The regression parameter, depending on both the MCP method and the selected reanalysis data set, strongly influences the outcome additionally.

     Note that the influence of the seasonality in $\beta_{1,LR}$ as shown in Fig. 3 (b) can not be determined exactly here, as the lower values in summer coincide with a stronger effect of the over-pronounced seasonal cycle of the reanalysis data (lower $d_{mean}$
values). However, the authors expect it to be rather small.

     In a study of Bass et al. (2000), long-term measurements instead of reanalyses were used as reference data. 41 pairs of site and reference data in Europe and the US with different terrain types were deployed to test a variety of MCP methods including linear models like linear regression as well as a neural network approach. Hence, long-term corrections of one-year on-site data were performed. Regarding the bias in mean wind speed they found that none of the investigated methods stood out in
comparison to the others. It was concluded that the success of the methods "is less to do with the mechanics of the methodology itself, and more to do with facets of the data being analysed". Carta et al. (2013) confirms that the uncertainty of the long-term predictions depends much more on the (reference) data than on the MCP method.

     With regard to an LTC of short-term wind measurements, the results of this work only partly agree with these findings. It was shown both theoretically and experimentally that, concerning systematic, seasonal biases, a strong dependence on the
selected MCP method occurs. The results above show that very different outcomes can be observed when relatively similar, linear MCP approaches are applied even when the same reference data set is used. In this context, it should be noted that when a long-term correction of entire one-year measurement data is performed, the seasonal aspects discussed here can be replaced by inter-annual variations (which are, however, much smaller); Eq. (13) retains its validity in this case.

     In a study of Weekes and Tomlin (2014a) seasonal patterns in the long-term correction of short-term wind measurements
are addressed briefly. For both LR and VR, larger biases in mean wind speed were observed when measuring in summer while smaller (more negative) values were obtained for winter measurements. The VR method yielded a smaller amplitude and, in contrast to the LR approach, resulted in negative biases throughout. Furthermore, it was concluded that the sign of the bias varied depending on the specific site when the VR method was applied.

     Weekes and Tomlin (2014a) related these seasonal effects to temporal changes in synoptic weather patterns and, connected
to that, seasonal patterns in wind direction. It has to be noted that Weekes and Tomlin (2014a) used measurements instead of





reanalysis data as reference and all data were collected at heights of around 10 to 20 m. The theoretical background derived in Sect. 4.2 is, however, independent of height and origin of the wind data and can be seen valid universally and applicable also under these conditions. From that it is likely that not all the reference data used in Weekes and Tomlin (2014a) exhibited an over-pronounced seasonal cycle as present for the reanalysis data used in the study here.

Saarnak et al. (2014) applied a linear regression approach to wind data from a site on a Swedish island using MERRA reanalysis (predecessor of MERRA-2). Systematic underestimations in a long-term correction were found when short-term data of three-months winter periods were used. Summer measurements, in turn, resulted in positive biases in mean wind speed. Hence, results similar to the ERA5 curve in Fig. 5 were obtained. Explanations for this seasonality were not given in the study.

### 5.4.2    Seasonal bias in variance

In this section, the bias of the MCP predictions with respect to variance is analyzed. Fig. 6 (a) and (b) show the respective error score $Err_{var}$.

The curves displayed in Fig. 6 (a) for the VR method resemble the inverse course of that observed in Fig. 4, thus, the patterns in the deviations in variance. This is not surprising, as the ratio of variances of measurement and reference data is used as a regression parameter in the VR method. Therefore, an error in the variance given by the reanalysis data has a strong impact on

$Err_{var}$. In summary, the theoretical analysis presented in Sect. 4.3 is confirmed by these experimental results. Connected to that, no clear overall seasonal course can be observed when the VR method is used. The amplitudes of the variations, however, are of distinct magnitude and remarkable errors can be observed.

As shown in Fig. 6 (b) a clear seasonal cycle of $Err_{var}$ is obtained when the LR method is applied. Lower values are present when measuring in summer and higher values can be found in case of winter measurements. This effect can be observed for

all reanalysis data sets. In Sect. 4.3 three parameters were identified which have a notable impact on $Err_{var}$ using the LR method. The authors suggest that the most important factor is the correlation coefficient as this parameter is known to exhibit a strong seasonal cycle and, above that, goes squared in the theoretical calculation of $Err_{var}$ (Eq. (16)). Fig. 7 underlines this assumption showing a clear seasonal variation of the correlation coefficient for all reanalysis data. Furthermore, this explains the substantial differences between Fig. 6 (a) and (b), i.e., between the results of the VR and the LR method.

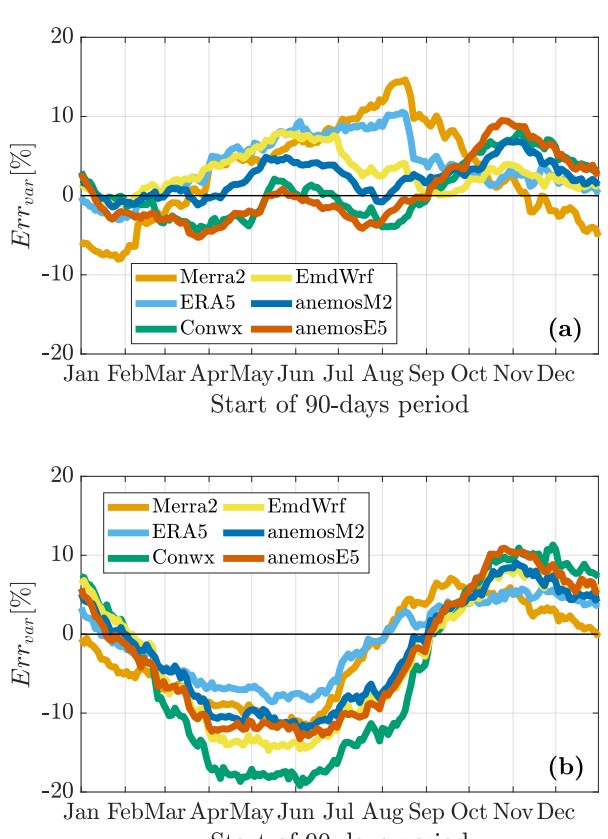

**Figure 6.** Seasonal bias in variance $Err_{var}$ using (a) Variance Ratio, (b) Linear Regression with Residuals method.

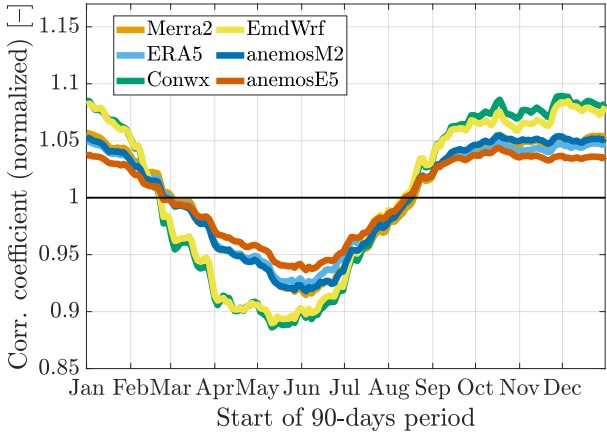

**Figure 7.** Normalized linear correlation coefficient between measurement and reanalysis data (periods of 90 days, arithmetically averaged over all sites). In the context of normalization the curves were shifted to a mean of 1 to better identify the (relative) temporal variations during the year.



In summary, the amplitudes in Fig. 6 (b) are generally of slightly larger magnitude than those of the variations produced by the VR method. This indicates that the VR method enables to obtain a more accurate variance of the corrected data on average. Differences occur regarding the type of reanalysis data. Similar to the bias in mean wind speed, ERA5 gives the lowest bias in variance when the LR method is used while large biases are obtained when the VR method is applied on the ERA5 data.

### 5.4.3    Seasonal bias in energy

In Sect. 4.1 a much higher importance of an accuracy in mean than in variance was obtained when aiming for a precise estimate of the energy in the wind. This contrasts with the finding of significantly higher biases in variance than in mean wind speed. In this section, the bias both in energy density ($Err_{ED}$) as well as in the theoretical energy production of a wind turbine ($Err_{turbine}$) is investigated based on experimental analysis. Emphasis is put on the eventual overall influence of $Err_{mean}$ and $Err_{var}$, respectively.

Figure 8 (a) and (b) show the error score $Err_{ED}$. The respective values were obtained according to Eq. (10) using the experimentally derived error values of $Err_{mean}$ and $Err_{var}$ as presented in Sect. 5.4.1 and 5.4.2, respectively. For $A$ the experimentally obtained average value of $A = 5.0$ was used (see Sect. 4.1). Hence, the diagram was produced by a weighted sum of $Err_{mean}$ and $Err_{var}$.

Additionally, the biases in $u^3$ based on the time-series values were evaluated experimentally (not shown here). These gave
very similar results to that presented in Fig. 8 (a) and (b) indicating that Eq. (8) contains a high validity despite the applied simplifications.

Comparison of Fig. 8 with the plots of $Err_{mean}$ and $Err_{var}$ (Fig. 5 and 6) reveals the influence of the biases in variance and mean wind speed on the bias in energy (density or production). Periods of contrary behaviour of $Err_{mean}$ and $Err_{var}$ (e.g., opposite sign or individual peaks) are most suitable to analyze this aspect here.

Generally, the influence of the bias in mean dominates (compare the considerations presented in Sect. 4.1). In some cases, however, the influence of the bias in variance is visible. E.g., in Fig. 8 (b) the sky-blue curve associated to the ERA5 data gives negative values in summer although the related $Err_{mean}$ curve remains positive in this period. This can be traced to the strongly negative $Err_{var}$ values here. When the VR method is used (Fig. 8 (a)), the effect of erroneous variance is even more clearly visible, due to different courses of the respective $Err_{mean}$ and $Err_{var}$ values, e.g., in case of the EMD-ConWx or the
anemosE5 data.

The bias in the theoretical energy production of a wind turbine $Err_{turbine}$ is shown directly below in Fig. 8 (c) and (d) allowing a good comparison of the two error scores. The courses of $Err_{turbine}$ show striking differences to the curves of $Err_{ED}$ indicating a large influence of the power curve on the respective results. Comparison with the patterns of $Err_{mean}$ and $Err_{var}$ reveals that the error in mean wind speed is even more decisive for the error in energy production than the theoretical
considerations suggest. The seasonal courses in $Err_{turbine}$ are very similar to the seasonal biases in mean wind speed $Err_{mean}$ (see Fig. 5). Its values are approximately twice the ones for the bias in mean wind speed. The influence of the bias in variance obviously is decreased by the power curve and barely visible. This is caused by the effect that variations of very large wind speed values exceeding the rated wind speed of the turbine contribute strongly to variance but do not affect the energy output.





However, in specific periods when $Err_{var}$ is large and its seasonal course does not follow the pattern of $Err_{mean}$, the influence

of $Err_{var}$ can be seen. Again, this is most clearly visible in case of the VR method (see, e.g., the data points related to the MERRA-2 or ERA5 data in the mid of August or to the anemosE5 data in fall in Fig. 8 c) in comparison to Fig. 5 a)).

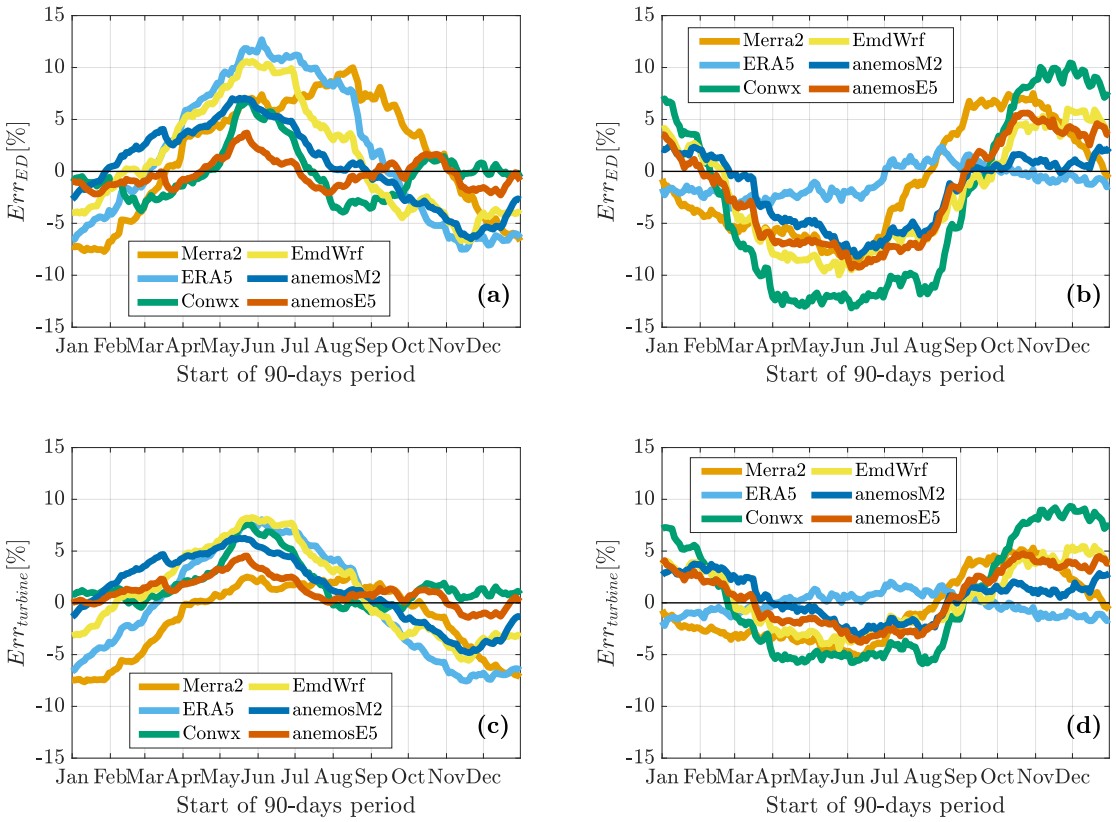

**Figure 8.** Seasonal bias in the prediction of the energy density $Err_{ED}$ ((a) and (b)) and the theoretical annual energy production of a wind turbine $Err_{turbine}$ ((c) and (d)). The figures on the left ((a) and (c)) refer to the VR method while in figures (b) and (d) the results produced by the LR method are shown.

A further difference between $Err_{ED}$ and $Err_{turbine}$ stands out when the VR method is applied. Some curves in Fig. 8 (c) mostly lie above or below zero for the entire year. Such "overall biases" are present especially in the case of the EMD-ConWx (positive overall bias) and the MERRA-2 data (negative overall bias). When applying the LR method (Fig. 8 (d)), hardly any

overall bias can be found.

Towards an explanation approach for these overall biases it should be noted that, again, the VR method produces higher values for the slope ($\beta_1$) than the LR approach. For the offset ($\beta_0$), the same formula is used in both MCP methods, relating offset to slope (see Eq. (3)). As a direct consequence, lower values for the offset are obtained when the VR method is applied.





For the VR method, hence, smaller wind speed values generally are corrected towards smaller values, while higher values are
increased compared to the correction applied in the LR method. This is visualized in the scatter plot in Fig. 9 where distinct
differences between the regression lines can be observed. Hence, wind speeds of small or rather large magnitude are corrected
differently. Similar correction is performed for wind speeds near the mean (i.e., values close to 1 in Fig. 9).

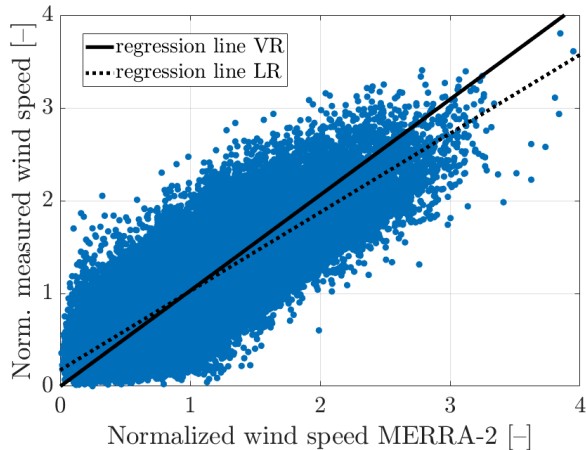

**Figure 9.** Scatter plot of normalized measured and MERRA-2 data and regression lines to these data using either the VR or the LR method.
Normalization was performed by dividing all wind speed values by the overall measured mean. The diagram was produced using the entire
measurement data of the 18 sites and the related MERRA-2 data.

This aspect can be expected to average out when considering mean wind speeds. However, it apparently becomes important
in case of energy production estimation where the cubic dependence on wind speed as well as the shape of the power curve lead
to a different importance (or weighting) of different wind speed values. Eventual wind-speed dependent errors of the reanalysis
data can further contribute to this issue.

The reasons for the overall biases, therefore, shows to be connected to characteristics of both the MCP method and the
reanalysis data set. Again, the combination of both these facets prove to be decisive with regard to the accuracy of an LTC.

### 5.4.4 Variation between the sites

Bias values of mean, variance or energy production should not be regarded as the only key figure to describe the accuracy of a
long-term correction procedure as it does not represent the overall uncertainty. In addition, the scatter, i.e., standard deviation
of the individual biases (in terms of variation between the sites) can be judged an important measure as it characterizes the
reliability of the results. Therefore, the standard deviation of $Err_{turbine}$ in dependence of the measurement period is briefly
addressed here and shown in Fig. 10. The analysis is restricted to $Err_{turbine}$ as this parameter is expected most useful for the
wind industry.

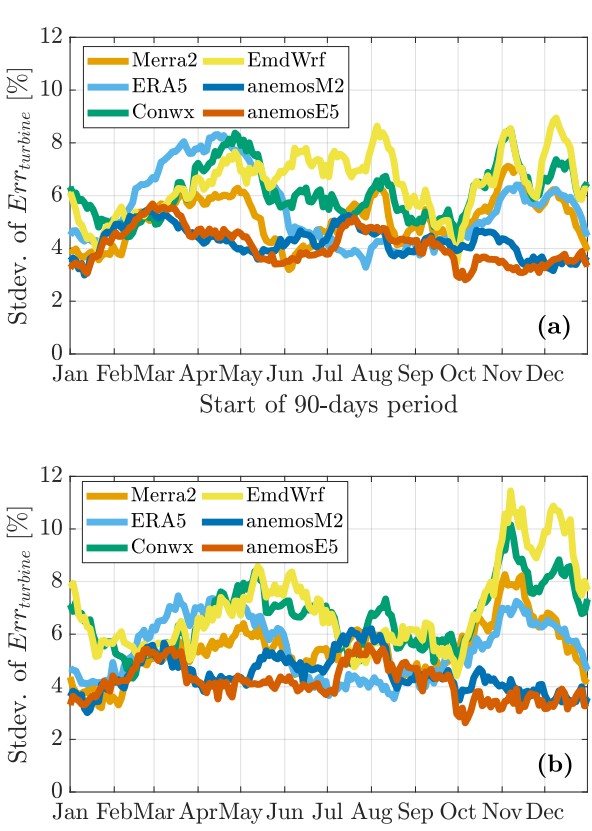

**Figure 10.** Bias variation between the sites (1 standard deviation) with regard to the accuracy of predicting the theoretical energy production of a wind turbine. (a) Variance Ratio, (b) Linear Regression with Residuals method.

The variations (standard deviations) are of comparable magnitude for both methods but, again, depend on the selected reanalysis data set. On average, smallest values can be observed in the beginning of the year and in fall (i.e., measurement period starting in January/February or September/October). This indicates that not only strong biases are present when the measurement is conducted in summer or winter but also higher variations, hence, smaller reliability of these biases can be

expected. Once more, this underlines the significance and importance of a sorrow selection of the measurement period, with transitional seasons (spring, fall) to be recommended in Central Europe.

The maximum values for individual reanalysis data sets in Fig. 10 range from almost 6 to 9 % in case of the VR method and from almost 6 to more than 11 % when the LR method is applied. The variation between the sites, therefore, is roughly of the same magnitude as the bias values themselves (see Fig. 8).





# 6 Conclusions and outlook

This study delivered in-depth analysis of seasonal effects in the long-term correction of short-term wind measurements. The provided findings can contribute to a further development of reanalysis data as well as improved MCP methods in this respect.

In a first step, the importance of the accuracy in mean and variance was analyzed with regard to a precise estimate of the energy in the wind. It was shown that the relative error in mean is weighted six times as strong as the relative error in variance in this context. Experimental analysis, in contrast, yielded much larger biases of the MCP predictions in variance than in mean (absolute values of more than 15 % in comparison to values of ± 4 %, respectively). Analyzing the biases in the theoretical energy production of a wind turbine showed that –apart from "overall biases"– the shape of the seasonal course of the bias in mean was more or less replicated here. It was concluded that the bias in variance should not be neglected, though; however, a much larger importance should be attached on a precise estimate of the mean.

A formula was derived which delivered the explanation for the seasonal biases in mean wind speed when applying either the VR or LR method. It was shown that the representativity of the measurement, hence, the similarity of the wind conditions of correlation and correction period, is important. Moreover, the capability of the reference data to reproduce the seasonal course is significant. Lastly, the regression parameter $\beta_1$ showed to be decisive for the magnitude of the seasonal biases.

These findings were confirmed experimentally. The largest biases were observed in non-representative wind conditions (i.e., summer or winter) with the magnitude depending on the reference data set. Furthermore, a strong dependence on the MCP method was determined; very different, partly even contrary characteristics in the seasonal biases were found for the VR and LR methods. In contrast to findings of existing publications, hence, this study showed that the biases in mean wind speed are connected to characteristics of both the reference data set as well as the MCP method.

For the error in energy production, in general, measurement periods in transitional seasons (spring, fall) not only resulted in smallest biases but also gave smallest variation between the sites, hence, the highest reliability of the results. The amplitudes of seasonal bias and standard deviation were roughly of same magnitude. Short-term wind measurements are, therefore, highly recommended to be conducted in periods of representative wind conditions (with respect to mean wind speed).

It should be noted, that the findings of the theoretical considerations can be seen valid independently of the chosen reference data as well as of the measurement duration. I.e., they are applicable on the case of a long-term correction of one-year wind data also.

Further research is necessary on how the systematic biases and, finally, the uncertainty can be reduced in an efficient and expedient way. The authors suggest that this could be approached in different ways. On the one hand, a manual correction based on the experiences above would reduce the biases. However, the reliability (standard deviation) would not change. A statistics-based approach (e.g., averaging the results of different MCP approaches and/or reference data) can be expected to result in larger improvements. On the other hand, the shortcomings of the reference (here: reanalysis) data in reproducing the seasonal course could be addressed. Discrepancies regarding temporal changes in synoptic weather patterns or atmospheric stability processes can be named as possible examples for such weaknesses. The inclusion of further meteorological data reflecting these characteristics could form the basis of a physically motivated approach here. The usefulness of removing seasonal biases



in e.g., wind profile extrapolation by including additional parameters like relative humidity was demonstrated in Basse et al.
(2020). This approach could also be taken here.

*Author contributions.*  AB had the lead in writing the manuscript and developing the theoretical analysis and methodology for this study. AB also performed all data analysis and visualization. LP contributed to the conceptualisation, development of the methodology, and to writing the manuscript. LP, DC, and AG had a supervisory role during the development of the methodology, data analysis, and the writing process. DC was also responsible for the funding acquisition and the project administration. AG performed valuable preliminary work. All authors
revised and edited the manuscript.

*Competing interests.*  The authors declare that they have no conflict of interest.

*Acknowledgements.*  The authors would like to express their gratitude to GWU Umwelttechnik GmbH, Notus Energy, NES GmbH, Meteorological Institute of the University of Hamburg, and Karlsruhe Institute of Technology for providing measurement data. Furthermore, the authors thank EMD Deutschland GbR and anemos GmbH for providing mesoscale reanalysis data.

*Financial support.*  This research was funded by the Federal Ministry of Economic Affairs and Energy (Bundesministerium für Wirtschaft und Energie, BMWi) on the basis of a decision by the German Bundestag, Grant No.: 0324159E.



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
