# Peer review of "Seasonal effects in the long-term correction of short-term wind measurements using reanalysis data"

_Wind Energy Science, 2020_

## Author Comment (AC1)

**Response to Anonymous Referee #1, wes-2020-134**

The authors would like to thank the referee very much for the extensive feedback and the numerous and valuable comments. We consider these very helpful and are sure that they will help to improve the quality of our paper. In the following, we present how we considered and incorporated the re-marks into the revised manuscript (responses in blue).

- - -

**General comments**
The preprint "Seasonal effects in the long-term correction of short-term wind measurements using reanalysis data" by Alexander Basse et al. deals with Measure-Correlate-Predict (MCP) methods in the context of a shorter-than-standard measurement periods (3 month instead of 1 year).
The long-term correction of in-situ measurements is an important step of any wind resource assess-ment for developing a wind farm. Thus, the topic is very relevant for the wind energy industry. In par-ticular, this focus on 3-month measurements could help make a better use of preliminary or incom-plete measurement campaigns.
Among the variety of MCP methods, the authors investigate 2: the Variance Ratio and the Linear Re-gression with residuals. This choice is relevant as they are amongst the most used MCP methods.
The question is addressed on a theoretical level and using a large set of wind-speed measurement data (18 sites) and a set of 6 relevant reanalyses as long-term reference data.
Despite those qualities, some flaws in the theoretical analysis impair the argumentation and conclu-sions. The major issue being that the role of the correlation between measurements and references is not enough considered nor investigated. The measurement data set is also under-exploited as site differences (arising from various terrains, various correlations to the reference data, or various meas-urement years) are not taken into account to analyse the results.
Sections 4 & 5 are too long, difficult to follow, for little conclusion. They could be written much more concisely. The reading is also impaired by grammatical mistakes and awkward phrasings.

Thank you for your appreciation of our paper as well as the general remarks, which we considered below in the responses on the specific comments.

**Specific comments**
**Title.** The title does not make clear that you are dealing with non-standard MCP, using only 3 months of data, not 1 year. Maybe rephrase it?

We agree that the title does not refer to 3-month periods. However, we assume that our main con-clusions are valid relatively independent of the measurement duration (we mention that in, e.g., lines 233-237 of the revised manuscript). Moreover, we feel that the term "seasonal" in the title already gives a sense of periods less than one year. We therefore would like keep the title as it is.

**§1 Introduction**
L25: I don't think that a "combination of observations and numerical models" is a fair definition of reanalysis.

We agree and changed the explanation of the reanalysis in the revised version to
*"Reanalysis data sets are produced using numerical weather simulations with a fixed state-of-the art model and assimilating historical weather data. In contrast to models used for weather prediction,*

*which are often updated and changed during operations, they therefore provide temporally consistent data sets over periods of up to several decades.”* (lines 25 ff. in the revised manuscript).

**§2 Data**
L80 "an entire year". Could you precise (in Table 1 or elsewhere) which year is available at each site? Are they all from the same year or from various years? You should also add some information about the average wind speed at each site and the correlation coefficient with each reference data set. All of this is important to understand the results.

In the description of Table 1 we write "The measurements were carried out between May 2013 and April 2019". We agree that more information on the measurement periods would be helpful. Unfortunately, we are not allowed to give further information on the exact period nor the wind conditions at the individual sites as this is commercially sensitive (data partly obtained from wind industry). The one-year periods are distributed relatively homogeneously between May 2013 and April 2019; only the year 2016 may be judged slightly over-represented (with eight of the 18 sites covering at least a few months of the year 2016). We added these information on the temporal distribution of the measurements in the revised manuscript (lines 83 ff.). Regarding the correlation coefficient we added some further information as well (see below, §3.3).

L93: MERRA-2 is also available as instantaneous values (at hourly resolution). Why not use this data set instead of the time-averaged one?

We think that it is consistent to use the time-averaged MERRA2 data and not the instantaneous values, since the measurements represent averaged values as well. Moreover, we face the scale difference between measurement and reanalysis data (point values versus grid cell data). This impairs comparison of spot values, while temporal averaging will help at least a little. Therefore, we preferred time-averaged reanalysis where available. We agree that investigating the instantaneous vs. averaged MERRA-2 data could be interesting as well, but is beyond the scope of this work and would not contribute to the main conclusions drawn from the analysis.

**§3 Methodology**
L128: you should state more clearly in this paragraph that, in this paper, ST=3 month and LT=1 year, while, usually, ST=1 year and LT=10-20 years. You should consider adding a graph or something to better explain your notations and the connexion between the various series (a. finding correlation between $u_{meas}$ and $u_{ref}$; b. predicting $U_{corr,LR/VR}$ from $U_{ref}$ ; c. verifying $U_{corr}$ against $U_{meas}$...).

We state the discrepancy to a "usual" LTC (ST=3 month and LT=1 year) in lines 142 ff. As recommended, we added a diagram for a better understanding of the notations and the procedure in the revised manuscript (Figure 1). We prefer to present these additional information after having explained the general procedure as we feel this facilitates the understanding for the reader.

§3.3
This section should contain the actual definition (formulas) of the scores. In fact, $Err_{mean}$ and $Err_{var}$ are never defined in the manuscript (except that $Err_{mean}$ is given L250-251, but called "theoretical" there, why?). And the figures are given in %: % of what? How do you normalize?

We agree and added the definitions at the respective passages. The error scores are calculated as relative values, e.g., $Err_{mean} = \frac{\bar{U}_{corr} - \bar{U}_{meas}}{\bar{U}_{meas}}$, which is why they are given in %. We expect that this becomes clearer now that the formulas are given in the manuscript. We call $Err_{mean,theo}$ "theoretical" in order to make clear that this factor represents a theoretical, mathematically derived parameter (in contrast to the experimentally derived errors shown in the diagrams).

[major issue] You apply the MCP methods to all sites and periods with all references. Is the correlation between reference and measurements high enough in all cases? Linear MCP methods should not be used if the correlation is low. If some combinaisons of [site, period, reanalysis] do not meet this criterium (with a sensitivity analysis on the correlation threshold), they should be removed from the results.

We calculated the correlation coefficient of all combinations of site, reanalysis data set and short-term period. In most cases, the correlation coefficient was > 0.65 throughout the whole year despite the small amount of only 90 days of data. We feel that showing all of these is too extensive, but we give some numbers in the revised manuscript (see lines 158 ff.). In general, this work explores the use of MCP methods and the implications of their usage under non-ideal conditions – i.e. measurement periods, which are much shorter than a year. Therefore, we think that quality indicators like correlation coefficients are not expected to match the standard criteria.
The correlation coefficient certainly is an important measure when considering long-term corrections. Therefore, the influence of the correlation coefficient is explored in the analysis. Our results suggest, however, that it is not the decisive criterion for deciding whether a long-term correction using short-term measurement data can be performed; instead, the (expected) accuracy should be used - i.e., the errors which are presented in the respective figures. However, the temporal variation of the correlation coefficient (as shown in Fig. 5) is crucial especially when linear regression is applied.

All results are shown averaged over all sites. First, it is not always clear how you average the sites. It should be detailed in this methodology section.

We agree and added some information on that in the methodology section (lines 153 ff.).

[major issue] Then, what about the variability across sites? You address it only at the very end of the manuscript (and not in a satisfactory way). But in the first figures, we have no idea whether the "average" behaviour is representative of all sites and if the differences between references are significant or not. There sould be some intervals associated with the mean curves.
(+previous comment on the need of maybe removing some sites).

We agree that more information on the variability would be useful. Unfortunately, adding intervals (error bars) to the mean curves would make the diagrams cluttered and confusing.
Therefore, we had decided to address the variability in a dedicated section. In the revised manuscript, we added the standard deviation (as a measure of variability across the sites) of $Err_{mean}$ in the respective figure (Fig. 11 in the revised manuscript). Furthermore, in lines 387-388 of the manuscript, we added a remark that analysis on the variability is presented in a later section.
We restricted the variability analysis on $Err_{mean}$ and $Err_{turbine}$ as we expect these parameters most useful for the wind industry. As shown and discussed, the error in variance generally can be regarded to be of minor importance for wind energy applications; the same applies to $Err_{ED}$. Hence, we feel

that showing all standard deviation values would make the paper too long with only little additional valuable information.

**§4 Theoretical considerations**

§4.1

This section is confused between the energy density (energy available in the wind, proportional to u$^3$) [let's call it ED] and the power production i.e. the output of a wind turbine (not proportional to u$^3$ because of the shape of the power curve) [let's call it PP]

- L221-222 "the wind power (…) target value of a WRA" . This is PP
- L224 "power in wind P": this is ED, not PP

We apologize for the confusion. We corrected the respective passages.

And since PP (what we want) is not proportional to u$^3$, I do not find this analysis very relevant. Using u$^3$ puts way too much weight over high wind speeds that finally do not produce power (>cut-out) or only the nominal power.

In this remark as well as in remark to § 5.4.3, the following issues are addressed
a) a lack of representativity when one power curve is selected (error score $Err_{\text{turbine}}$)
b) a lack of relevance for the wind industry when $u^3$ is considered (error score $Err_{\text{ED}}$).

On aspect a): As suggested below, we tried two further power curves (significantly lower and higher rated wind speeds than the power curve being used in the study so far). The results differed slightly with regard to the amplitude of $Err_{\text{turbine}}$, but the essential conclusions remained the same. We added a remark on that in the revised manuscript (lines 228-231 in Sect. 3.3).
On aspect b): We agree that the accuracy in energy density is not of primary interest for the wind industry. However, its strength lies in the general validity (independent of a power curve). Therefore, we decided to use
1) $Err_{\text{ED}}$ as a power-curve independent error score for the energy in the wind (moreover, $Err_{\text{ED}}$ might be interesting for different aspects of research, if not for industry)
2) $Err_{\text{turbine}}$ as an error score which depends on the selected power curve but shows a high relevance for wind industry. We used the same power curve also in an earlier publication (see https://doi.org/10.3390/rs12071091) successfully and therefore think it is advantageous to keep 1) and 2).

§4.2

Eq 13: I do not find this decomposition so relevant. It is difficult to interpret because it is a difference, not a sum, of 2 terms.

The goal of this formula was to reveal the different influencing factors on the bias in mean wind speed when a long-term correction of short-term wind measurements is performed. In our opinion, it delivers the mathematical background for understanding the seasonal biases. We are not convinced that a sum would make it easier to interpret than a difference. In contrast, we would expect that the difference allows a better understanding that both terms act in different directions.

§4.3

L282-284: I am not sure that this is the right conclusion.

If we want $\text{var}(U_{\text{corr}}) = \text{var}(U_{\text{meas}})$, Eq 15 gives: $\text{var}(u_{\text{meas}})/\text{var}(U_{\text{meas}}) = \text{var}(u_{\text{ref}})/\text{var}(U_{\text{ref}})$

Hence, what is important is that the ratio ST-variance over LT-variance should be similar in the reference and the measurement. So, if a reference is always under/over-dispersive, it could OK (regarding this particular aspect). Also note how this relates to the parameter $d_{\text{var}}$ introduced later in §5.3.

We agree that the whole passage is a bit misleading and, therefore, rephrased it in the revised version. Furthermore, we changed the order of the terms in Eq. 15 (Eq. 14 in the revised manuscript) to

$$\text{Var}(U_{\text{corr}}) = \text{Var}(u_{\text{meas}}) \cdot \frac{\text{Var}(U_{\text{ref}})}{\text{Var}(u_{\text{ref}})},$$

hoping that this further facilitates understanding the conclusion.

L290-292: You did not explain how the residual distribution is fit, i.e. how $\sigma_\varepsilon$ relates to the other properties (and I could not access the reference paper). Hence some questions:

- Is is possible to really distinguish between effects 2 & 3?
- Isn't effect 3 important and worth investigating? You do not mention it ever again.

We added some information on how $\sigma_\varepsilon$ is calculated (lines 194-198). Furthermore, we agree that $r_{\text{ref,meas}}$ and $\sigma_\varepsilon$ are strongly connected (higher scatter around the fitting line generally results in lower correlation and higher $\sigma_\varepsilon$ values) and can/should not be treated independently. We added a remark on that in the respective passage. Because of this relation between $r_{\text{ref,meas}}$ and $\sigma_\varepsilon$, we restrict the analysis on effects 1. and 2. (lines 310-313 in the revised manuscript).

**§ 5 Experimental results**

§ 5.1 to § 5.4.1

L313: it is unclear why you introduce this particular parameter $d_{\text{mean}}$, not directly related to the previous theoretical considerations.

In fact, you could have linked $d_{\text{mean}}$ to $\text{Err}_{\text{mean}}$. If you go further from Eq (13), you would see that:

$\text{Err}_{\text{mean}} = \text{mean}(U_{\text{corr}}) - \text{mean}(U_{\text{meas}}) = -\boldsymbol{\beta_1} \, U_{\text{ref}} \, d_{\text{mean}} + \ldots$

This explains what you see in the experimental section.

From our perspective, it is useful to introduce a figure, which enables to identify differences in the seasonal courses of reanalysis and measurement data. In our opinion, this figure can only be defined in a sensible way by either a difference or a ratio of wind speeds in short and long-term period.

Hence, as an alternative to our approach it would be possible to define $d_{\text{mean}}$ as

$$d_{\text{mean}} = \frac{\bar{u}_{\text{ref}} - \bar{U}_{\text{ref}}}{\bar{u}_{\text{meas}} - \bar{U}_{\text{meas}}}$$

In this case, Eq. 13 could be transformed to contain $d_{\text{mean}}$ as suggested in the comment. For the original definition in the paper ($d_{\text{mean}} = \frac{\bar{u}_{\text{ref}}}{\bar{U}_{\text{ref}}} - \frac{\bar{u}_{\text{meas}}}{\bar{U}_{\text{meas}}}$), rewriting Eq. 13 as $\text{Err}_{\text{mean}} = \text{mean}(U_{\text{corr}}) - \text{mean}(U_{\text{meas}}) = -\boldsymbol{\beta_1} \, U_{\text{ref}} \, d_{\text{mean}} + \ldots$ is not easily possible.

However, the formulation of $d_{\mathrm{mean}}$ above brings other problems. In periods of similar wind speeds in short and long-term period (i.e., $\bar{u}_{\mathrm{meas}} \approx \bar{U}_{\mathrm{meas}}$), the denominator takes values close to 0 and $d_{\mathrm{mean}}$ becomes very large (positive or negative). This cannot be depicted in a diagram appropriately, which is why we had decided against this option.

Therefore, we decided to introduce $d_{\mathrm{mean}}$ as a measure for presenting and intuitively understanding the differences in the seasonal courses of reanalysis and measurement data - despite the fact, that it cannot directly be included in the theoretical analysis in a sensible way.

The same applies for the parameter $d_{\mathrm{var}}$ accordingly.

L330-341 (§ 5.2): it feels like you are discovering that $\beta_{1,\,\mathrm{VR}} > \beta_{1,\,\mathrm{LR}}$ while this is central and should be obvious from the start.

We rephrased the text in the revised manuscript to avoid this impression.

L345: again it is unclear why you introduce this particular parameter $d_{\mathrm{var}}$, and how it relates to the theory in §4.3.

Please see our response above regarding $d_{\mathrm{mean}}$.

L381: of course the differences between LR and VR arise from $\beta_1$. This should be stated much earlier and investigated thouroughly.
Considering that $\beta_{1,\,\mathrm{LR}} = r^2\, \beta_{1,\,\mathrm{VR}}$ , the correlation coefficient is very important and should be considered way before §5.4.2

We agree and therefore had addressed these differences in $\beta_1$ in a distinct section (Sect. 5.2). In the revised manuscript, we moved the parts regarding the correlation coefficient to this (earlier) section.

You should also try to understand the differences between seasons and sites originating from $\beta_1$:

- $\sigma_{\mathrm{meas}}/\sigma_{\mathrm{ref}}$ : whether the reference is under/over dispersive: how does this vary spatially, temporally and among reanalyses?
- [for LR only]: $r^2$ : how does the correlation vary across the sites and across the year? Why is the correlation lower in spring? Is it a general conclusion or is it linked to one/some particular(s) year(s)? (Are all the measurements from the same year or very different years? cf. Data section)

The temporal variation as well as the variation among the reanalyses is shown in Fig. 3 (a) ($\beta_1$ equals $\frac{\sigma_{\mathrm{meas}}}{\sigma_{\mathrm{ref}}}$) and Fig. 4 (in the revised manuscript: Fig. 4 (a) and Fig. 6). We added a corresponding remark in Sect. 5.3 to make this clearer to the reader (lines 371-372).

We decided to exclude the discussion about the spatial variation of the statistics $\beta_{1,\mathrm{LR}}$, $\beta_{1,\mathrm{VR}}$, and the correlation coefficient as it would extensively increase the scope of the paper instead of keeping the focus on the systematic biases in an LTC. Furthermore, our data basis of 18 sites is not sufficient to derive reliable results when trying to separate the influences of seasonality, spatial variation and different years. We therefore restrict the analysis here to average behavior.

Regarding the correlation coefficient, please see our comments above (§2 and §3.3). Additionally, we added a remark on possible reasons for the lower correlation for spring/summer periods (atmospheric stability, lower wind speeds). We certainly think that the conclusion can be drawn generally for Central Europe / Germany as the measurement periods originate from different years and sites (see comment above, §2).

L406-431: some literature review that might be better in the introduction?

We agree that this literature review would also fit well in the introduction. On the other hand, we feel that a direct comparison of the results obtained in Sect. 5.4.1 with results from literature is helpful for the reader to judge the validity of the results at this point. We therefore left this passage in this paragraph.

§ 5.4.3
For the "ED" score: why not use mean($u^3$) directly?

We use Eq. (10) for calculating $Err_{\mathrm{ED}}$ in order to directly present the influences of $Err_{\mathrm{mean}}$ and $Err_{\mathrm{var}}$ on $Err_{\mathrm{ED}}$ and to compare the results to $Err_{\mathrm{turbine}}$.
As we write in lines 464-466 (lines 494-496 in the revised manuscript), we tested mean($u^3$) directly. We decided against showing both (mean($u^3$) and $Err_{\mathrm{ED}}$ from Eq. (8)) for means of the length of the paper.

And relevance for the industry (cf. §4.1)?
The results of the "turbine" score may depend a lot on the turbine's choice (especially its rated wind speed, here 14 m/s, and, to a smaller extent, its cut-in wind speed). Did you conduct any sensitivity analysis? If not, you should try several power curves having different rated wind speeds.

Please see our response above (§4.1).

**§6 Conclusions**
L541: "Short-term wind measurements are recommended to be conducted in periods of representative wind conditions", but the whole point of a MCP method is to correct the fact that the ST period is not representative of the LT!
In practice,

- how would you know that a ST period is representative enough?
- given the very high inter-annual variability in wind speed, even if the ST period is in spring or fall, it does not seem that you could guarantee that they would be close to the LT mean, does it?

We admit that the question of "representative periods" is rather opened than answered. Of course, you cannot know in advance what a representative period really is (nor guarantee representativity). However, from our study we learned that transitional seasons (spring, fall) are most likely to give the best results while showing representative wind conditions (in terms of mean wind speeds similar to the average mean). We rephrased the respective passage accordingly.

Have you considered the possibility of non-contiguous 3-month periods? For example if a LIDAR is moved around 3 or 4 sites, changing place every month?

We agree that this would be an interesting aspect to analyze with a good chance to gain more accurate estimates (while, on the other hand, too extensive for this paper, we assume). Additionally, from our experience in practice, splitting the (already short) measurement periods in two campaigns is not very attractive for the wind industry (probably because of the relatively high effort and costs for moving the lidar).

**Technical corrections**
Maybe use different notations for residuals in Eqs 1 and 6.

We accounted for that in the revised manuscript.

Subscript labels (meas, ref, LR, …) should be typeset upright, not italicized (use \mathrm{} in LaTeX). Idem for mathematical function "var". And "Err" in italics is weird, maybe use a symbol instead, e.g. δ or Δ?

We considered this in the revised manuscript. We prefer "*Err*" versus δ or Δ as the latter are commonly used for (absolute) differences, not for relative deviations.

L182: you could say "zero mean" instead of "mean μ" (confusing)

We considered this in the revised manuscript.

L345 (Eq 18): the subscripts say "tar" instead of "meas"

We corrected this in the revised version, thank you.

English:

- Revise the use of "which": missing commas (e.g. L 28, 45, 144-145, ...) ; improper structures (e.g. "relationship are derived which trace..." L8, "studies have been presented which ..." L47, "is given which" L153 …)
- Revise the position of frequency adverbs, which should be placed after a form of the verb to be ("often" (L 56, 144), "generally" (L 28, 111, 154, 494), "usually" (L 140, 251) ...)
- Revise the use and positioning of "therefore"
- Revise punctuation. Consider adding some commas: some are absolutely necessary (e.g. L33 before "providing", L180 before "giving" etc.) ; some would be useful in long sentences (e.g. L144 around "in practical applications", L147 after "For… periods" etc.)
- L44: round-robin (no capital, I think)
- L105: time inconsistency "have replaced (…) did"
- L114 and later: "annual/seasonal course": wouldn't "seasonal cycle" be a better term?
- L135, 305 etc.: 90-day periods (not 90-days)
- L187: the VR method is proposed as … (verb position)
- L235-236: rephrase (a value is not "present" at a site)
- L251, 257 and other places: when speaking of bias, maybe use terms such as
- "difference" instead of "deviation"
- L259: remove ",hence,"
- L279: redundancy "respective (…) respectively"

- L279 and later: uncommon abbreviations "cmp." and 's.': maybe use "cf." or "see"
- (…)

Thank you very much for this comprehensive list of language improvements, which we really appreciate and considered when revising the manuscript.

---

## Author Comment (AC2)

**Response to Anonymous Referee #2, wes-2020-134**

The authors would like to thank the referee very much for the extensive review. We think that the remarks are very helpful and that they will help to improve the quality of our paper. In the revised manuscript, we considered the comments as follows (responses in blue).

- - -

Summary: This papers aims to test two MCP (measure-correlate-predict) methods for a series of re-analysis products and analyze whether errors in the mean or in the variance contribute most to the total error or not. The paper is an interesting analysis and presents useful insights in the math behind the MCP methods. The paper is however difficult to read and needs major revisions.

Recommendation: major revision required

Thank you very much for the positive general feedback.

Major remarks:

1. My main concern is that the paper is not easy to read. Many parts of the text have very short paragraphs of only one or two sentences, so here and there the text is very fragmented.

    We have removed several line breaks, rephrased some sentences and made some linguistic improvements.

    Also the paper does not provide sufficient explanation of the parameters used. The further I got into the paper, the more often I had to go back to the beginning of the paper to find out what was exactly meant by different terminology. So more physical/dynamical meaning should be given to the parameters that are introduced.

    We have added some additional explanations in the revised manuscript. As we feel that too many repetitions would raise other problems, we had decided to give a general explanation of the subscript labels at the beginning of the methodology section (lines 129-132 in the revised manuscript). Considering this general explanation, we hope that the parameters used in the paper are now easier to understand. Additionally, we have worked with some references to equations and sections in order to guide the reader to the respective definitions if necessary.

2. The paper concerns only the MCP methods for mean wind and energy (u^3). But I was wondering whether extreme events in the wind field like ramps (up and down), low level jets and wind shears would also be interested to study, since they have a big impact on the wind energy production variability in time and on the wind turbine installations.

    Long-term correction is a procedural step in the preparation of wind resource assessments, which is described in various guidelines (e.g., FGW e.V. 2020; MEASNET 2016). It is our aim to publish relevant findings for this step of the procedure and we would like to focus precisely on this. Despite the mentioned aspects would certainly be interesting to investigate, we therefore decided not to include these in this study.

Extreme events do not play a central role for the statistical estimation of the wind speed distribution in Germany on land and, thus, the long-term wind resource. Therefore, wind ramps do not lie in the focus of our study. However, they are to some extent implicitly included in other parameters we investigate (e.g., variance of wind speed, or $Err_{\mathrm{ED}}$).

Similarly, we did not include analysis on low level jets as they do not play a significant role at onshore sites in Germany.

The use of the rotor-equivalent wind speed would also make the seasonal variation of the shear interesting to study. However, this would be beyond the scope of this paper and should be addressed separately.

3. In terms of re-analyses that were used in the study I was surprised the COSMO REA family (https://reanalysis.meteo.uni-bonn.de/?Download_Data___COSMO-REA2) was not present, since it was especially made for Germany.

   We discussed taking these data into account, because both the REA 2 and the REA 6 data would in principle be very interesting for Germany due to their COSMO assimilation procedure. However, the global data (ERA-Interim) driving the DWD REA data already expired in 2019. This means that the data has no significance for long-term references of current and future wind farm projects and was therefore not taken into account by us. Besides that, the COSMO-REA2 data are only available for the period of 2007-2013 (according to the link you provide). The measurement campaigns from which we used the data for the analysis were carried out later than this period.

4. It is a pity that the paper only use the different re-analysis products as illustration for their mathematical exercise. I think for many WES readers it would be interesting to be more specific under which meteorological conditions which product "is best" or "performs less accurate". Also it would be interesting whether more detail can be added about the physics/dynamics behind the variability. Is the error due to missed sea breezes, or Alpine pumping events or low level jets etc. The observational data across Germany is very rich so more of this kind of info might be extractable.

   In our study, we investigate the theoretical and empirical background of the seasonal effects (including reasons). We agree that a broader examination of under which meteorological condition which reanalysis data perform best, would be interesting. However, we feel that enlarging the study in this way would go beyond the scope of a clear and focused paper. Similarly, we agree that regional differences would be interesting to study, but too extensive to be included as part of this publication.

5. The Conclusion section should be rewritten since I find there is too much jargon in it. Conclusions are based on the beta parameters, but this makes the conclusion difficult to read as a separate text, which many people do. Please reword.

   We revised the conclusion, rephrased some sentences and added some information in order to make this section easier to understand even when not having read the whole paper.

Smaller remarks:

Ln 19: please cite in chronological order, here and please check complete manuscript.

We changed the order of the references accordingly in the revised manuscript.

Ln 45: overperform: do you mean "outperform"?

Yes, thank you for that remark. We changed it in the revised manuscript.

Ln 55: Strange sentence: if the costs are so low, it is an argument to do more rather than less experiments.

We rephrased that sentence and added a remark on the running costs of lidar measurements.

Ln 95: Better to refer to the Hersbach 2020 paper in QJRMS.

We changed the reference in the revised manuscript.

Ln 134-135: Maybe I misunderstand the strategy here, but if you have taken 90 day periods with each 3 days intervals, then you still sample from a complete year (I read it as i you take 1 Jan, 4 Jan, 7 Jan .....). So this is not how a measurement campaign occurs where maybe only one or two months are sampled.

Indeed, we take 1 Jan, 4 Jan, etc. as starting points for the 90-day periods. We added this information in the revised manuscript in order to make that clear. Consequently, we have 122 individual (while overlapping) periods and each period is investigated independently from the others. In this way, it is possible to calculate average statistics (e.g., errors in mean wind speed) for these 122 90-day periods. These form the basis for the question which errors occur (on average), when a measurement campaign is started on a certain date.

Ln 134-135: if you complete the series at the end of the series with the new year, is the winter overrepresented in this analysis?

The winter is not overrepresented, as each 90-day period of a full year is considered exactly once in the analysis (see explanation above). For the case that a 90-day period exceeds the period in which measurement data is available, the data from the beginning of the measurement year is appended.

Example:

Measurement data at site X is available for Aug 2018 - Jul 2019 (i.e., the "real" measurement campaign as part of our data basis took place at that time). For the 90-day period covering the summer (or rather: the 90-day period June 1 - Aug. 29), the data from June and July 2019 and from August 2018 are used.

Of course, this does not correspond to the procedure in practice. It can be used in the analysis, however, as the correlation (MCP) is not restricted to contiguous measurement periods.

Ln 170: please explain more early in the manuscript what are umeas, Umeas, uref and Uref.

As mentioned above, we explain the general meaning at the beginning of the methodology section (lines 129-132 in the revised manuscript).

Ln 202: a one-year time series is generated: but this is inconsistent with was written in line 134-135 where you say you sample 90-days periods.

The measurement data from the 90-day periods is correlated with the reanalysis data and MCP predictions are performed. The result is a one-year time series (i.e., the reanalysis data is corrected for a period of 1 year). In the methodology section, we added a diagram in order to further explain the procedure.

Ln 217: extent

We corrected that, thank you.

Ln 216-219: this paragraph is extremely abstract

We have rephrased it in the revised manuscript.

Ln 257: explain why "true" is between "".

As the sentence reads "Deviation of "true" mean wind conditions (measured data) in measurement and long-term period", we feel that the supplement "(measured data)" explains what we mean with "true" mean wind conditions. We use quotation marks as the measured wind data does not exactly reflect the real wind speeds (because of measurement errors), but is expected to be accurate enough in this study to identify errors of the MCP predictions.

Ln 260: representativity: do you mean representativeness??? Please check several places in the manuscript.

We have changed that in the entire manuscript.

Ln 312: Differences occur in the amplitudes.: short and weird sentence. What do you want to say?

We rephrased that to "The amplitudes of the curves (…) differ, indicating clear differences between the reanalysis data sets." (line 333).

Figure 1: please add in the caption how the normalization was done, so the reader does not need to go through the manuscript again to look it up.

We have done so in the revised manuscript.

Ln 326: might be caused .....: this is speculative. Please prove what you would like to say here.

As we did not get detailed information on the developments done by anemos, this certainly is speculative (despite quite likely, though). In this passage, we find it reasonable to mention the differences in the two data sets nevertheless.

Ln 330: Please explain more what you want the reader to learn from Fig 3.

We added an explaining remark on that at the respective passage (Sect. 5.2).

Figure 5: caption: please reword caption. You do not show seasonal bias, but the bias through the different months. The plot does not show bias for DJF, MAM, JJA, SON...

We changed it to "Temporal variation during the year of the bias in mean wind speed (…)." Additionally, we changed the captions of the figures regarding the other error scores (variance, energy) accordingly.

Ln 391: "or rather because of the erroneous seasonal course of the ERA5 data.": this is not clear to me since Fig 1 says that ERA5 has a correct seasonal cycle.

We rephrased the sentence using the term "overpronounced annual cycle" in order to make it easier to understand.

Ln 405: However, the authors expect it to be rather small: argue why, prove with data or physical reasoning.

We removed this sentence in the revised manuscript.

Ln 413: these: please indicate to what "these" refer to

We added a remark making clear that we refer to the findings from literature (lines 443-444).

Ln 428: From that it is likely that not all the reference: messy sentence that makes the reader lost.

In the revised manuscript, we rephrased that sentence (lines 458-459).

Ln 446: "The authors" -> we. Now it sounds as if you place it beyond yourself.

We rephrased that in the revised manuscript to "It can be expected that…".

Figure 9: the variables on the x and y axes should be switched, since the observation is the known and the MERRA is the modelled/predicted.

Since the observation is the target variable and MERRA-2 is the input variable (of the MCP), we feel that the variables should not be switched.

- - -

**References**

FGW e.V.: Fördergesellschaft Windenergie und andere dezentrale Energien (FGW): Technical Guidelines for Wind Turbines: Determination of Wind Potential an Energy Yield (TR6)., 2020.

MEASNET: Measuring Network of Wind Energy Institutes: Evaluation of Site-Specific Wind Conditions: Version 2 April 2016, 2016.

---

## Author Response (AR2)

**Response to the reviewers, wes-2020-134**

The authors would like to thank the associate editor and the referees once more for the review and the helpful remarks. We are certain that they helped to further improve the quality of our paper. In the revised manuscript, we considered the comments as follows (responses in blue).

- - -

**Associate Editor (Sara C. Pryor)**

As you will see from the reviewers comments - they perceive - and I concur that the changes made do not fully address their concerns and the manuscript remains rather difficult to follow. I'd therefore like to request you seek to modify your manuscript in line with their suggestions - I would particularly emphasize the need to clarify and edit to reduce length in sections 4 & 5. I would also ask you to revisit the theoretical considerations and how they are described in section 5. If you can do so I will make a final determination without further review.

In accordance to your recommendations, in this revision we focused strongly on clarifying and shortening sections 4 and 5. The changes consist of the following (the line numbers refer to the revised manuscript):

1.) We rephrased several sentences such as in lines 283, 286, 306-307 as well as entire passages, e.g., in the beginning of Sect. 5.1 (please see track-changes file for all changes made).

2.) In some cases, we added further information in order to clarify the descriptions and explanations (e.g., lines 341-343, 346).

3.) In the former version of our manuscript, we had often justified the individual approaches in the subsections of Sect. 5 (mainly with references to the theoretical findings, which may have made things a bit confusing). We now summarized a general motivation at the beginning of Sect. 5 ("In the theoretical analysis, different factors were identified which have an impact on the accuracy in mean and variance …"). We believe that this approach is more straight-forward and expedient.

In this context, the structure of some sections was changed to 1.) description and explanation of the observations (figures), 2.) reasons and consequences of the results. Also, the length of the sections was reduced.

4.) We shortened sections 4.2, 5.2, 5.3 (see track-changes file). Moreover, we reduced Sect. 5.4.3 to the bias in theoretical energy production of a wind turbine. Thus, in this context, we decided to leave out the error score $Err_{\mathrm{ED}}$, i.e., the bias in energy density. This was motivated by our impression that $Err_{\mathrm{ED}}$ caused quite a bit of confusion with only little added value to the main findings of the paper.

Overall, despite additional explanations, we were able to shorten the manuscript by 29 lines (equivalent to almost one page of plain text). Moreover, we expect that both the theoretical and the experimental results will now be much easier to read and understand. With all the changes made, we hope that the paper now meets your expectations.

**Anonymous Referee #1**

Minor corrections to be made:
L261: "the questions which factors"
L298: "the ratio of the variances given by the reanalysis data" unclear
L530: "Eventual" ?
L532&574: "both, "
L561: "much" -> more
L566: "significant" ?
and probably some other English problems but I would leave that to a native English speaker.

As displayed in the track-changes file, all these technical corrections have been considered and implemented.

**Anonymous Referee #2**

Major remarks:

1. Abstract: The abstract elaborates quite a lot on the activities that has been employed during the study. However, the abstract should report about the conclusions and findings of the paper, i.e. what have we learnt?. Now that is covered in only a single sentence.

We revised the abstract and added the required information about the essential findings of the paper.

2. Readability: Section 3.1 as an example: From the beginning this section is difficult to follow. I think some additional explanation is needed that explains the research strategy in a more concrete way. E.g. the first sentence "Short-term periods with a duration of 90 consecutive days are selected starting at the first day of year and running through the data with an increment of three days" does not mentioned that you do this operation on the reanalysis products, so for the reader it is a bit shaky here: where are we suddenly heading to?

Moreover, so far often wordings like "analyse" and "statistics are generated" are used, but for the reader it is difficult to follow why this is exactly done. Perhaps a flowchart that explains the design of the research would be helpful.

We revised Sect. 3.1, especially the first passage, in order to clarify the procedure. In our view, this section is now structured in a way that makes the approach of the study easier to understand. We now explain more clearly 1.) how the selection of data in the short-term periods is done, 2.) in what way the data in these periods are analyzed, 3.) how the MCP predictions are performed. Furthermore, we changed Figure 1 accordingly. On the one hand, this scheme now better captures the procedure of MCP prediction, on the other hand, the use of capital and lower case letters as well as the subscripts ($u_{\mathrm{meas}}$, $U_{\mathrm{corr}}$ etc.) should become clearer.

Also the text can be more precise in terms of using the word "data" which is used for the observations and for the re-analysis products, which makes the reader easily lost what is exactly

done. For example, in ln 136 "ensuring that 122 90-day measurement periods can be investigated", the measurement periods that are mentioned are subsets of the reanalyses, not from the observations in Table 1. So this makes the paper confusing. I suggest to use the words "observations" and "re-analysis product", or if you sample from a reanalysis to mimic a measurement campaign, please call this subset "pseudo observations" (or something like that).

We did not use a sample from a reanalysis to mimic a measurement campaign but understand that the explanation was a bit misleading. We hope that our changes made here (e.g., "When this sliding window reaches the end of the period of the original measurement campaign …", "This procedure is applied equally to measurement and reanalysis data…", lines 145 - 148) will help to make the passage easier to understand.

In addition, in ln 139 you talk about regression parameters but at this point the reader has no concrete idea about which regression parameters you talk about concretely! It remains all a bit vague in a cloud. In general I have confidence in the work the authors present, but less confusing wordings can help to make the paper more attractive for the readership, which will help to have a bigger impact in the field.

We sharpened the passage you refer to (see lines 153 - 158 in the revised manuscript). In addition to the revision of Fig. 1, we hope that these changes will now make the general approach taken in the study much clearer.

3. Discussion: the paper should sharpen the discussion section in which the paper shows how it has extended the science. The paper now refers continuously to three papers, but I wonder whether that can be strengthened? The reference list contains a lot of grey literature, the paper can be brought to a higher scientific level.

It is not entirely clear to us which discussion section you are referring to, but we assume that the relevant passages in Sect. 5 (especially Sect. 5.4.1) are meant. We added a further passage on how the paper adds value to existing publications in lines 455-458 (in order to account for these changes, the conclusion was slightly adjusted accordingly).

Furthermore, we conducted a further literature search which, however, did not reveal any additional studies outside of grey literature on this topic. Therefore, we expect that all relevant journal papers have been considered. (Please also see the papers cited in the introduction which, however, did not provide further analysis on seasonal biases.) Again, this shows that there is a lack of reviewed publications. With our paper, we aim to contribute to this issue. The large amount of grey literature might be explained by the application proximity of the research. Industrial research unfortunately tends not to publish in journals.

Minor remarks:

Ln 32: MCP: the MCP method should be explained briefly or with a schematic (see below as well), since from section 3.1 the manuscript is difficult to follow, which can be circumvented by including a flowchart or other schematic about how the MCP works (inputs, output, procedures, regressions; so an extension of Figure 1), and a flowchart illustrating the experiment. From Fig 1 it is still not clear what is the difference between a u and capitalized U.

We have taken up this suggestion of a schematic by revising Fig. 1 (see above). We believe that this scheme better reflects both the procedure of MCP predictions and the use of upper and lower case

letters. In addition, the general explanation of the use of upper/lower case letters (and subscripts) in lines 139 - 141 as well as several brief notes when the respective parameters are used (e.g., lines 185, 200, 288, 291) should make the definitions of the parameters clearer to the reader.

Ln 41: scientific publications -> studies
Ln 41: In Carta et al. (2013) an extensive review is given on -> Carta et al. (2013) presents an extensive review on
Ln 42: It is concluded that -> They concluded that....
Ln 78: An overview of the measurement campaigns is given in Tab. 1. -> Table 1 presents an overview of the measurement campaigns used in this study.
Ln 102: WRF (2020))-> the usual reference to the WRF model is either Powers et al 2017 (BAMS) or Skamarock et al 2019.
Ln 132: I recommend to replace the labelling of the subscript "ref" with the more direct subscript "reanalysis". I understand in general other reference wind speeds can be used as well, but for the readability of the paper I think it is wise to make the terminology as direct as possible.
Ln 138: three-month -> to avoid confusion, just call it "90-day periods", so the wording remains consistent with the previous paragraph.

We have fully considered these recommendations and implemented them in the revised manuscript accordingly.

Ln 138: In a first step, the data in these three-month data portions 90-day periods are analyzed with respect to, e.g., mean and variance of wind speed. For the reader it is unclear why these statistics must be generated at this point. Again, adding a flowchart or a scheme in the beginning of section 3.1 can help to make the workflow more easy to understand for the reader.

We clarified this passage as stated above. E.g., we added specific reasons for why these statistics are generated and investigated (see lines 150 - 152 of the revised manuscript). In order to avoid confusion, we restricted the flowchart (Fig. 1) to the procedure of MCP predictions.

Ln 153: few negative wind speed values can occur. -> You mean that the MCP method may generate a few negative wind speed values. Be more precise in the wording, it will help the reader!

We changed the wording here accordingly.

Figure 1: the location and role of the grey bars "benchmark Umeas" are unclear. The grey bars next to the dark blue bar now look purely like they act to refine the layout. Also, there are no black arrows pointing to/from it. Please revise.

We revised it as stated above.

Ln 166: reference wind speed. For readability purposes, it would be good that uref refers to the reanalysis products in this case right?

We changed the index "ref" to "rea" (for reanalysis) consistently.

Ln 173: please add that the subscript LR stands for linear regression

We added this remark in lines 198 - 199.

Ln 193: what is the difference between Eq 4 and Eq. 6 since they effectively do the same job? Is it not more confusing to mention them twice?

The main difference of these two equations is that in one case the residuals are considered, while in the other case they are not. While this does not change the mean, it does have a significant effect on the variance. Therefore, although the equations look similar, they produce quite different results. We therefore would like to keep both equations in order to illustrate this difference between simple linear regression and linear regression with residuals.

Ln 215: one-year time series: please be more precise. It is a daily, hourly or sub-hourly time series?

We added the information on the temporal resolution (line 234).

Ln 254: Inserting in Eq. (10). What are you inserting in Eq. 10? A is already present in that Equation.

We added the missing information ("Inserting this mean value of A in Eq. (10)…" in line 270).

Ln 265: suddenly overbars appear above the U's in the equation, but without further explanation. What was the averaging timescale? Unclear (again).

We added "the bar denotes the mean" once more in line 280. The parameters which are used in Eq. (11) - (13) were already defined several times before (Fig. 1, caption of Fig. 1, lines 185 and 200), with the definitions directly containing the respective timescales (long-term / short-term period). Note that Eq. (11) - (13) are valid independently of the exact duration of short-term or long-term period. We added that remark in line 297. Together with the general explanation on lower/upper case letters in the beginning of Sect. 3, we believe that this section should be clear now.

Ln 335: the authors should introduce better why d_mean is needed. It is well explained what it does and how you can calculate it, but why do we need it? Which research question does it answer?

In order to further clarify the purpose of $d_{\mathrm{mean}}$, we added "It therefore indicates the average "error" of the reanalysis data sets in reflecting the annual course of wind speed. According to the theoretical considerations in Sect. 4.2, this is an important aspect regarding the seasonal biases of an LTC." in lines 341-342. Furthermore, we noted one result in lines 353-354 ("the largest amplitude prevails for the EMD-WRF Europe+ data set").

Figure 10: the caption should mention the height of the wind speed that is discussed

We added the required information in the form of a reference to Tab. 1 and Sect. 2, where all heights are specified.

Ln 561: much -> much more?

Yes, thank you, we changed that in the revised manuscript (line 534).